# Transcriptome and Proteome Analyses Revealed Differences in JEV-Infected PK-15 Cells in Response to Ferroptosis Agonists and Antagonists

**DOI:** 10.3390/ani14233516

**Published:** 2024-12-05

**Authors:** Xiaolong Zhou, Yiwei Chen, Xinyao Kang, Ayong Zhao, Songbai Yang

**Affiliations:** Key Laboratory of Applied Technology on Green-Eco-Healthy Animal Husbandry of Zhejiang Province, College of Animal Science and Technology, College of Veterinary Medicine, Zhejiang A&F University, 666 Wusu Road, Hangzhou 311300, China; zhouxiaolong@zafu.edu.cn (X.Z.); yiweiya159@163.com (Y.C.);

**Keywords:** ferroptosis, transcriptome, proteome, JEV, PK-15 cell

## Abstract

Japanese encephalitis virus (JEV) is a mosquito-borne neurotropic flavivirus that causes acute viral encephalitis in animals. Ferroptosis occurred in encephalitis, but the relationship between JEV and ferroptosis is still unclear. Ferroptosis, an iron-dependent and regulated form of cell death driven by excessive lipid peroxidation, has been shown in previous studies to play an important role in controlling the proliferation of viruses. This study investigates the effects of ferroptosis on the proliferation of JEV in the PK-15 cells. Transcriptome and proteome sequencing identified differentially expressed genes and proteins in response to treatment with ferroptosis agonists or antagonists. The results offer a more comprehensive understanding of the molecular mechanisms by which ferroptosis influences JEV in PK-15 cells.

## 1. Introduction

Japanese encephalitis virus (JEV) is a mosquito-borne neurotropic flavivirus that can cause epidemic encephalitis B. JEV is a single-stranded RNA virus with neurotoxic and neuroinvasive characteristics. Functional proteins encoded during JEV replication include the NS3 protein, which has three distinct enzyme activities that contribute to viral replication and assembly. Infection with JEV results in acute central nervous system diseases in humans. JEV infection outcomes in pigs do not vary by breed. JEV infection occurs frequently in pigs in summer. JEV infection in pigs can lead to miscarriage in sows, while in boars, it may result in orchitis, testicular shrinkage, and hardening or loss of spermatogenic function, eventually leading to infertility. Additionally, piglets may die from JEV-induced encephalitis [1,2,3].

Ferroptosis is a form of regulated cell death that has gained significant attention in recent years. Unlike traditional forms of cell death, such as apoptosis, necrosis, and autophagy, ferroptosis is distinct in its biochemical characteristics and regulatory pathways [4,5]. It is primarily driven by the accumulation of reactive oxygen species (ROS), particularly lipid peroxides, which leads to cell membrane damage and ultimately cell death. Iron plays a crucial role in ferroptosis, as excess intracellular iron can catalyze the formation of ROS through the Fenton reaction. This reaction generates highly reactive hydroxyl radicals, which initiate lipid peroxidation [6]. Antioxidant defense systems play a critical role in protecting cells from ferroptosis. Glutathione (GSH), a major cellular antioxidant, is essential for maintaining the redox balance. During ferroptosis, the depletion of GSH leads to an increase in lipid peroxidation [7]. The enzyme glutathione peroxidase 4 (GPX4) uses GSH as a cofactor to reduce lipid peroxides and prevent ferroptosis. Inhibition of GPX4 activity can trigger ferroptosis, making it a key target for inducing this form of cell death in cancer cells [8]. Previous studies have shown that viral infections decrease GSH levels and reduce the activity of GPX4, both of which are critical for preventing ferroptosis [9,10]. Inhibition of the GPX4–GSH antioxidant system is a key factor in ferroptosis induction [11,12]. Viral infections can also impair the system Xc–GSH–GPX4 axis, triggering ferroptosis as a way to evade the immune response [12]. Viruses disrupt the cellular redox environment, leading to excessive ROS generation through the Fenton reaction, which relies on the accumulation of iron [9,13]. This results in oxidative stress, lipid peroxidation, and ultimately ferroptosis [13]. The transcription factor NFE2L2 plays a protective role by mitigating oxidative stress and iron overload, regulating antioxidant, detoxification, and iron metabolism pathways [14]. GCLC is an important subunit of glutathione synthetase, which participates in the synthesis of glutathione (GSH) together with GCLM. GCLC can enhance the antioxidant capacity and detoxification ability of cells, protect cells from oxidative stress and harmful substances, and maintain cell homeostasis and survival through promoting GSH synthesis [15,16]. GPX4—a critical regulator of ferroptosis—reduces lipid peroxides in membranes, maintaining membrane integrity, and regulating iron metabolism through reducing lipid peroxides in the membrane [17]. GPX4 can regulate iron metabolism by binding to iron ions and preventing the Fenton reaction [17,18]. NQO1 plays a crucial role in detoxifying lipid peroxyl radicals and inhibiting ferroptosis [19,20,21,22]. PTGS2, also known as COX-2, is widely recognized as a marker for ferroptosis [23,24]. Several signaling pathways are involved in the regulation of ferroptosis. The transcription factor nuclear factor erythroid 2-related factor 2 (NRF2) plays a protective role against ferroptosis by upregulating the expression of antioxidant genes [25,26]. Conversely, the p53 tumor suppressor protein can promote ferroptosis under certain conditions. Additionally, the activation of the endoplasmic reticulum stress response and the unfolded protein response can also contribute to the induction of ferroptosis [27,28,29]. Ferroptosis is generally considered a stimulatory factor for viral proliferation. Certain viruses, such as CVB3, IAV, MMTV, and CPT, can bind to TFR1 to enter cells, leading to iron accumulation and ferroptosis [12,30], which in turn promotes viral replication. JEV and HSV have also been demonstrated to be able to induce ferroptosis through inhibiting the GSH-GPX4 system, ACSL4 phosphorylation, and Nrf2 ubiquitination [31]. However, the exact functions of ferroptosis in relation to viruses remain unclear. Further research is needed to fully elucidate the mechanisms of ferroptosis in the context of viral infections and to develop effective therapeutic interventions based on this knowledge.

In this study, we aimed to explore the role of ferroptosis in terms of the proliferation of JEV. To this end, JEV-infected PK-15 cells were treated with ferroptosis agonists or antagonists to evaluate their potential impacts on JEV proliferation. As a porcine kidney epithelial cell line, PK-15 cells have been shown to exhibit similar susceptibility and function as skin epithelial cells, making them a valuable model for JEV research. Furthermore, many studies have already examined JEV infection using PK-15 cells. Thus, PK-15 cells represent a robust model to investigate the role of ferroptosis in the host response to JEV infection. We found that the ferroptosis antagonist promoted the proliferation of JEV, whereas the ferroptosis agonist suppressed it. Transcriptome and proteomic analyses were performed to determine the pathways and regulatory networks involved in ferroptosis and its effects on JEV.

## 2. Materials and Methods

### 2.1. Cell Culture and Viral Infection

PK-15 cells were cultured in MEM containing 10% FBS and a 100 μg/mL penicillin/streptomycin mixture at 37 °C with 5% CO_2_. Antagonists and agonists were synthesized by Med Chem Express-MCE (Monmouth Junction, NJ, USA). We used SP600125 (HY-12041, MCE, Monmouth Junction, NJ, USA) and Fer-1 (HY-100579, MCE, Monmouth Junction, NJ, USA) as the ferroptosis antagonists; RSL3 (HY-100218A, MCE, Monmouth Junction, NJ, USA) and Erastin (HY-15763, MCE, Monmouth Junction, NJ, USA) were used as the ferroptosis agonists. The JEV strain SA14-14-2 (GenBank accession number: AF315119.1) was propagated in BHK-21 cells, and then the PK-15 cells were incubated in the JEV solution for 2 h at an MOI = 1; the inoculum was removed, the cells were washed three times with PBS, and fresh medium was added. Infected PK-15 cells were then cultured in MEM medium containing 2% FBS without penicillin/streptomycin.

### 2.2. Real-Time Quantitative PCR Analysis

Primers for *NFE2L2*, *GCLC*, *GCLM*, *GPX4*, *NQO1*, and *PTGS2* were designed using Primer 5 software. The *ACTB* (*β-Actin*) primer was used as an internal control. Total RNA was extracted from cells using TRIzol^®^Reagent (Invitrogen, Waltham, MA, USA) according to the manufacturer’s protocol. Total RNA (1 μg) was reverse transcribed in a 10 μL reaction volume using the RevertAid™RT kit (RR036A, Takara, Kusatsu, Shiga, Japan). The sequences of primers can be found in (Appendix A).

### 2.3. Western Blot Analysis

Cells were lysed using RIPA lysis buffer (P0013B, Beyotime, Shanghai, China), 1 mM PMSF (ST506, Beyotime, Shanghai, China), and a protease and phosphatase inhibitor mixture (P1050, Beyotime, Shanghai, China). The protein concentration of cell lysates was determined by the BCA protein assay kit (Pierce, Rockford, IL, USA). Ten micrograms of total protein from each sample were loaded onto sodium dodecyl sulfate polyacrylamide gel electrophoresis (SDS-PAGE) at 80 V for 3–4 h and then transferred to PVDF membranes (3010040001, Roche, Boston, MA, USA) using electroblotting at 300 mA for 90 min. After incubation for 1 h in blocking buffer (PBST containing 1% (*w*/*v*) BSA (A7030, Sigma, St. Louis, MO, USA), each band was cut according to the marker (NS3 60-75kD, TBB5 45-60kD) from the PVDF membrane and incubated with the primary antibody individually, and then membranes were incubated with NS3 (Cat: GTX125868, Genetex, Irvine, CA, USA). Following application of the primary antibody, the membranes were washed, and then horseradish peroxidase (HRP)-labeled mouse anti-rabbit IgG secondary antibody (Cat: sc-2357, Santa Cruz, CA, USA) or HRP-labeled mouse IgG Fc secondary antibody (Cat: sc-525409, Santa Cruz, CA, USA) was added for 1 h at room temperature and washed again. The membrane was visualized using the ECL Western blot detection kit (Cat: NC15080, Thermo, Norristown, PA, USA). We also examined the protein level of TBB5 (Cat: AM1031A, Abgent, Shanghai, China) as an internal control. Chemiluminescence intensity of each protein band was quantified using ImageJ software (V1.8.0, NIH, Bethesda, MD, USA), and protein levels were then normalized by the amount of TBB5 protein.

### 2.4. Oxidative Stress Analysis

The lactate dehydrogenase detection kit (C0016, Beyotime, Shanghai, China), lipid peroxidation MDA detection kit (S0131S, Beyotime, Shanghai, China), and reactive oxygen species detection kit (CA1410, Solarbio, Beijing, China) were used to analyze the levels of lactate dehydrogenase (LDH), malondialdehyde (MDA), and reactive oxygen species (ROS) in PK15 cells after JEV infection.

### 2.5. Transcriptome Sequencing and Differentially Expressed Gene Analysis

Total RNA from cells was extracted using TRIzol^®^ Reagent (Invitrogen) according to the manufacturer’s protocol. The concentration and purity of the proposed RNA were detected by Nanodrop2000, the integrity of the RNA was detected by agarose gel electrophoresis, and the RQN value was determined by Agilent5300. The RNA samples were commissioned to perform transcriptome sequencing by Shanghai Meiji Biomedical Technology Co., Ltd. (Shanghai, China). Reference genome for https://mart.ensembl.org/Sus_scrofa/Info/Index (accessed on 10 May 2024 ).

Fragmentation buffer was used to fragment the RNA, reverse transcription was used to synthesize cDNA, end repair, addition of A tail and adapter, amplification of DNA library, and quality detection. Clean reads were obtained by removing unqualified reads from the original data and aligning them to the reference genome. The expression levels of genes and transcripts were quantified separately using the software RSEM (V1.3.3, Berkeley, CA, USA). Genes with foldchange ≥ 2 and adjusted *p*-value < 0.05 between samples were identified as differentially expressed genes. GO (gene ontology) functional cluster analysis and KEGG (Kyoto encyclopedia of genes and genomes) pathway enrichment analysis were performed.

### 2.6. Proteome Sequencing and Analysis of Differentially Expressed Proteins

The BCA method was used to determine the protein content. Sodium dodecyl sulfate polyacrylamide gel electrophoresis (SDS-PAGE) was used to detect the protein quality. After a qualified protein quality test, Shanghai Meiji Biomedical Technology Co., Ltd. was commissioned to perform proteomic sequencing. The main steps were protein concentration determination, trypsin digestion, peptide quantification, chromatographic separation, liquid chromatography-tandem mass spectrometry analysis, and data analysis. Proteins with foldchange ≥ 2 and adjusted *p*-value < 0.05 between samples were identified as differentially expressed proteins, and then GO functional cluster analysis and KEGG pathway enrichment analysis were performed. All raw data were deposited in the CNCB GSA database (accession number: OMIX007500).

### 2.7. Statistical Analysis

Data are presented as mean ± SEM. The student’s *t*-tests or one-way analysis of variance (ANOVA) were performed using SPSS software (ver, 20.0, SPAA Inc., Fort Collins, CO, USA). For RT-qPCR data, *p* < 0.05 was considered statistically significant (* *p* < 0.05, ** *p* < 0.01). We conducted protein–protein interaction analysis using String (v11.5), which was then visualized using Cytoscape (v3.10.1) to generate a protein–protein interaction network map.

## 3. Results

### 3.1. Ferroptosis-Related Genes and Indicators Were Regulated by JEV Infection in PK-15 Cells

We found the MOI = 1, and incubating the JEV solution for 2 h is a suitable treatment for JEV infection in PK-15 cells. The mRNA and protein levels of JEV can be detected at 12, 24, 36, and 48 h; the cells didn’t initiate cytopathic effect (CPE), and at 60 h after infection, the cells initiate CPE. 

To investigate the influence of JEV on ferroptosis-related genes in PK-15 cells, RT-qPCR was performed. The results indicated that the expressions of the genes *NFE2L2*, *GCLC*, *GCLM*, *NQO1*, and *GPX4* were significantly downregulated after JEV infection (Figure 1A–E), whereas the gene *PTGS2* significantly increased (Figure 1F). Additionally, to assess the effects of JEV on ferroptosis-related indicators, the levels of LDH, ROS, and MDA were measured. The results showed that JEV infection significantly elevated the levels of LDH, ROS, and MDA in PK-15 cells (Figure 1G–I). Therefore, we consider that JEV infection can trigger ferroptosis in PK-15 cells.

### 3.2. Effects of Ferroptosis Agonists and Antagonists on JEV Proliferation

To investigate the effects of ferroptosis on JEV proliferation in PK-15 cells. Western blot analyses were performed. The results indicated that the ferroptosis antagonist Fer-1 slightly increased the protein level of JEV NS3 (Figure 2A), whereas the ferroptosis antagonist SP600125 obviously increased the protein level of JEV NS3 (Figure 2B). However, the protein levels of JEV NS3 were obviously decreased after treatment with the ferroptosis agonists Erastin and RSL3 (Figure 2C,D). RT-qPCR was also performed, and the results indicate that ferroptosis agonists Erastin and RSL3 significantly decreased the mRNA level of JEV, with Erastin showing a stronger inhibitory effect on both JEV mRNA and NS3 protein than RSL3 (Figure 2C–E). Furthermore, the mRNA level of JEV was significantly increased after treatment with the ferroptosis antagonist SP600125, but not with Fer-1 (Figure 2F). These results indicated that SP600125 and Erastin are more suitable for further experiments to study the regulatory effect of ferroptosis on JEV.

### 3.3. Differentially Expressed Genes Induced by Ferroptosis Agonist and Antagonist in JEV Infected PK-15 Cells

We performed RNA sequencing of JEV-infected PK-15 cells treated with the ferroptosis agonist or antagonist in order to investigate the differences in mRNA between the two treatments (*n* = 3) (Appendix A). The negative control (NC) group was JEV-infected PK-15 cells. All raw data were deposited into the NCBI sequence read archive (SRA) database (Accession Number: PRJNA1160171). In the RNA-Seq results, we obtained approximately 134.61 billion raw bases (891.4 million raw reads) and 132.47 billion clean bases (884.5 million clean reads) in total. The clean data were aligned to the Sus_scrofa Sscrofa11.1 (GCA_000003025.6) reference genome, with a mapping rate ranging from 96.51% to 97.14%. We detected a total of 26,786 genes, comprising 25,721 known genes and 1065 novel genes. The Q20 (the percentage of bases with sequencing quality above 98.92% in total bases) and Q30 of each sample were above 96.49%, indicating high-quality sequencing data that were suitable for subsequent analysis.

We identified 89 up-regulated and 35 down-regulated differentially expressed genes (DEGs) at 24 h in the SP600125 treatment group compared to the negative control (NC) group (Figure 3A). After 48 h of treatment, we identified 21 up-regulated and 4 down-regulated DEGs in the SP600125 treatment group compared to the NC group (Figure 3B). A total of 310 up-regulated and 202 down-regulated DEGs were identified at 24 h in the Erastin treatment group (Figure 3C). After 48 h of treatment, 17 up-regulated and 23 down-regulated DEGs were identified in the Erastin treatment group (Figure 3D). In contrast to the SP600125 treatment group, we found 113 up-regulated and 270 down-regulated DEGs at 24 h in the Erastin treatment group (Figure 3E), and after 48 h of treatment we observed 19 up-regulated and 74 down-regulated DEGs in the Erastin treatment group (Figure 3F). As JEV-infected PK-15 cells began to present cytopathic effects at approximately 48–60 h, the number of DEGs was significantly lower than at 24 h. Multi-group difference analysis of DEGs was also performed, and the expression trends of some DEGs in 24 h and 48 h were similar; many DEGs exhibited a different expression trend at 48 h compared with 24 h (Appendix A).

Notably, there were overlapping DEGs among the comparisons between different groups after treatment with the ferroptosis agonist and antagonist, with 45 genes being commonly differentially expressed between Erastin vs. NC and SP600125 vs. NC. As the expression trends of the overlapping genes were uniform, we focused on the non-overlapping DEGs between the groups. We found 467 DEGs between the Erastin and NC groups at 24 h after Erastin treatment. A total of 339 unique DEGs were found between the Erastin and NC groups at 24 h; these DEGs may contribute to the Erastin-induced ferroptosis in PK-15 cells and inhibit the proliferation of JEV. Additionally, 44 unique DEGs were found between the SP600125 and NC groups at 24 h after SP600125 treatment (Appendix A); these DEGs may contribute to the SP600125-suppressed ferroptosis in PK-15 cells, thus promoting the proliferation of JEV.

EggNOG classification indicated that, compared to the NC group, the DEGs were mainly enriched in posttranslational modification, intracellular trafficking, secretion, and vesicular transport at 24 h after SP600125 treatment. Similarly, the DEGs were mainly enriched in posttranslational modification, intracellular trafficking, secretion, and vesicular transport at 24 h after Erastin treatment; compared to the SP600125 group, the DEGs in the Erastin treatment group were also enriched in posttranslational modification, intracellular trafficking, secretion, and vesicular transport at 24 h (Appendix A). The results indicated that many DEGs corresponded with the Erastin and SP600125 treatments.

### 3.4. Gene Expression Pattern Analysis for Ferroptosis Agonist and Antagonist Treatment in JEV-Infected PK-15 Cells

GO enrichment analysis indicated that, compared to the NC group, the DEGs were mainly enriched in the regulation of immune response, defense response, and protein binding after SP600125 treatment (Appendix A). After Erastin treatment, the DEGs were mainly enriched in immune system process, defense response, and regulation of signal transduction (Appendix A). Additionally, compared to the SP600125 group, the results indicated that the DEGs were mainly enriched in the regulation of immune response, positive regulation of signal transduction, and regulation of inflammatory response after Erastin treatment (Appendix A).

The KEGG enrichment analysis was also performed. Compared to the NC group, the results indicated that the DEGs were mainly enriched in cytokine–cytokine receptor interaction, viral protein interaction with cytokines, the IL-17 signaling pathway, the TNF signaling pathway, the Toll-like receptor signaling pathway, and the chemokine signaling pathway at 24 h after SP600125 treatment (Figure 4A), and the TNF signaling pathway, IL-17 signaling pathway, and NF-kappa B pathway at 48 h (Figure 4B). Meanwhile, the DEGs were mainly enriched in viral protein interaction with cytokine, cytokine–cytokine receptor interaction, the Toll-like receptor signaling pathway, ribosome, and oxidative phosphorylation at 24 h after Erastin treatment (Figure 4C) and the TNF signaling pathway and IL-17 signaling pathway at 48 h (Figure 4D). Compared to the SP600125 group, the results indicated that the DEGs were mainly enriched in viral protein interaction with cytokine, viral carcinogenesis, cytokine–cytokine receptor interaction, and the chemokine signaling pathway at 24 h after Erastin treatment (Figure 4E) and the TNF signaling pathway and the IL-17 signaling pathway at 48 h (Figure 4F). The details of genes assessed in the GO enrichment and KEGG enrichment analyses are provided in Appendix A.

We also performed KEGG enrichment analyses on up- or downregulated DEGs, respectively. Compared to the NC group, the upregulated DEGs were mainly enriched in cytokine–cytokine receptor interaction, viral protein interaction with cytokines, the IL-17 signaling pathway, the Toll-like receptor signaling pathway, and the JAK–STAT signaling pathway at 24 h after SP600125 treatment (Appendix A); whereas, the downregulated DEGs were mainly enriched in the IL-17 signaling pathway at 24 h after SP600125 treatment (Appendix A). Compared to the NC group, the upregulated DEGs were mainly enriched in cytokine–cytokine receptor interaction, ferroptosis, the Toll-like receptor signaling pathway, and the P53 signaling pathway at 24 h after Erastin treatment (Appendix A); whereas, the downregulated DEGs were mainly enriched in viral protein interaction with cytokines at 24 h after Erastin treatment (Appendix A). Compared to the SP600125 group, the results indicated that the upregulated DEGs were mainly enriched in viral protein interaction with cytokine, cytokine–cytokine receptor interaction, and the Toll-like receptor signaling pathway at 24 h after Erastin treatment (Appendix A); however, the downregulated DEGs were mainly enriched in viral protein interaction with cytokine, cytokine-cytokine receptor interaction, the Toll-like receptor signaling pathway, the HIF signaling pathway, and the PI3K–AKT signaling pathway at 24 h after Erastin treatment (Appendix A). Compared to the NC group, the upregulated DEGs were mainly enriched in cytokine–cytokine receptor interaction, the TNF signaling pathway, and the IL-17 signaling pathway at 48 h after SP600125 treatment (Appendix A); whereas, the downregulated DEGs were mainly enriched in oxidative phosphorylation at 48 h after SP600125 treatment (Appendix A). Compared to the NC group, the upregulated DEGs were mainly enriched in steroid hormone biosynthesis at 48 h after Erastin treatment (Appendix A), whereas the downregulated DEGs were mainly enriched in viral protein interaction with cytokines, the TNF signaling pathway, the IL-17 signaling pathway, and cytokine–cytokine receptor interaction at 48 h after Erastin treatment (Appendix A). Compared to the SP600125 group, the results indicated that the upregulated DEGs were mainly enriched in motor proteins at 48 h after Erastin treatment (Appendix A); however, the downregulated DEGs were mainly enriched in viral protein interaction with cytokines, the TNF signaling pathway, the IL-17 signaling pathway, and cytokine–cytokine receptor interaction in the JAK–STAT signaling pathway at 48 h after Erastin treatment (Appendix A). Taken together, we found that upregulated DEGs were involved in Erastin-induced ferroptosis but not downregulated DEGs, and the IL-17 signaling pathway was regulated by both upregulated DEGs and downregulated DEGs.

To identify potential trends among the groups, we performed cluster analysis on all DEGs, which were divided into eight subclusters based on gene expression (Figure 5A). Since the effects of SP600125 and Erastin on ferroptosis are contrary, we focused on genes that exhibited different expression patterns between the SP600125 and Erastin groups. We found subcluster_1, subcluster_2, subcluster_5, subcluster_6, subcluster_7, and subcluster_8 to fit this expression pattern (Figure 5B). GO enrichment analysis of subcluster_1 indicated that the DEGs were mainly enriched in response to stress, cytokine activity, and structural molecule activity (Figure 5C). KEGG enrichment analysis of subcluster_1 indicated that the DEGs were mainly enriched in viral protein interaction with cytokine, cytokine–cytokine receptor interaction, the P53 signaling pathway, the Toll-like receptor signaling pathway, and the IL-17 signaling pathway (Figure 5D).

GO enrichment analysis of subcluster_2 indicated that the DEGs were mainly enriched in protein binding and cell projection (Figure 5E). KEGG enrichment analysis of subcluster_2 indicated that the DEGs were primarily enriched in complement and coagulation cascade reaction, the HIF-1 signaling pathway, the P53 signaling pathway, the Toll-like receptor signaling pathway, and the PI3K–AKT signaling pathway (Figure 5F).

### 3.5. Differentially Expressed Proteins Induced by Ferroptosis Agonist and Antagonist in JEV-Infected PK-15 Cells

An orbitrap astral high-resolution mass spectrometer was used to determine the qualitative and quantitative characteristics of protein profiles. We performed data-independent acquisition (DIA) proteomics on JEV-infected PK-15 cells treated with the ferroptosis agonist and antagonist to investigate the differences in proteins between the treatments (*n* = 3). The negative control (NC) group was JEV-infected PK-15 cells. All raw data were deposited in the CNCB GSA database (accession number: OMIX007500). In the proteomics results, we obtained 108,858 peptide fragments and 6381 proteins. The protein sequences were aligned to the databases of EggNOG, GO, KEGG, NR, Pfam, String, and Uniprot databases, with a mapping rate of 99.44%, 98.12%, 100%, 94.08%, 90.13%, and 100%, respectively.

After treatment with the ferroptosis agonist and antagonist, compared to the NC group, we identified 161 up-regulated and 149 down-regulated differentially expressed proteins (DEPs) at 24 h in the SP600125 treatment group (Figure 6A). After 48 h of treatment, compared to the NC group, we identified 120 up-regulated and 189 down-regulated DEPs in the SP600125 treatment group (Figure 6B). A total of 239 up-regulated and 317 down-regulated DEPs were identified at 24 h in the Erastin treatment group (Figure 6C). After 48 h of treatment, 250 up-regulated and 440 down-regulated DEPs were identified in the Erastin treatment group (Figure 6D). In contrast to the SP600125 treatment group, we found 261 up-regulated and 311 down-regulated DEPs in the Erastin treatment group (Figure 6E), and after 48 h of treatment, we found 308 up-regulated and 405 down-regulated DEPs in the Erastin treatment group (Figure 6F). Multi-group difference analysis of DEPs was performed, and while the expression trends of some DEPs at 24 h and 48 h were similar, many DEPs exhibited a different expression trend at 48 h compared with 24 h (Appendix A).

Notably, we found 425 and 565 DEPs between the Erastin and NC groups at 24 h and 48 h, respectively. A total of 201 and 217 unique DEPs were found between the Erastin and NC groups at 24 h and 48 h, respectively. After Erastin treatment, these DEPs may contribute to Erastin-induced PK-15 cell ferroptosis and inhibit the proliferation of JEV. A total of 102 and 108 unique DEPs were found between the SP600125 and NC groups at 24 h and 48 h, respectively. After SP600125 treatment, these DEPs may contribute to SP600125-suppressed PK-15 cell ferroptosis, thus promoting the proliferation of JEV (Appendix A).

EggNOG classification indicated that, compared to the NC group, the DEPs were mainly enriched in transcription, posttranslational modification, and intracellular trafficking at 24 h and 48 h after SP600125 treatment. Similarly, the DEPs were mainly enriched in transcription, posttranslational modification, and intracellular trafficking at both 24 h and 48 h after Erastin treatment. Compared to the SP600125 group, the DEPs were also enriched in transcription, posttranslational modification, and intracellular trafficking at 24 h and 48 h after Erastin treatment (Appendix A).

### 3.6. Protein Expression Pattern Analysis for Ferroptosis Agonist and Antagonist Treatments in JEV-Infected PK-15 Cells

GO enrichment analysis was performed, and compared to the NC group, the results indicated that the DEPs were mainly enriched in transmembrane signaling receptor activity, molecular transducer activity, positive regulation of apoptotic process, and positive regulation of programmed cell death at 24 h after SP600125 treatment (Appendix A). At 48 h after SP600125 treatment, the DEPs were mainly enriched in extracellular space, cell adhesion, and external encapsulating structure (Appendix A). For the Erastin treatment, the DEPs were mainly enriched in oxidoreductase activity, carboxylic acid metabolic process, and small molecule metabolic process at 24 h after Erastin treatment (Appendix A), while at 48 h, the DEPs were mainly enriched in small molecule metabolic process, organic acid metabolic process, oxoacid metabolic process, carboxylic acid metabolic process, and immune system process (Appendix A). Compared to the SP600125 group, the results indicated that the DEPs were mainly enriched in membrane at 24 h after Erastin treatment (Appendix A) and in defense response and immune response at 48 h after Erastin treatment (Appendix A).

KEGG enrichment analysis was also performed. Compared to the NC group, the DEPs were mainly enriched in mineral absorption, cytokine–cytokine receptor interaction, the TNF signaling pathway, ECM–receptor interaction, and focal adhesion at 24 h after SP600125 treatment (Figure 7A). At 48 h after SP600125 treatment, the DEPs were mainly enriched in ECM–receptor interaction, focal adhesion, and the PI3K–AKT signaling pathway (Figure 7B). For the Erastin treatment at 24 h, the DEPs were mainly enriched in ECM–receptor interaction, ferroptosis, cysteine and methionine metabolism, the PI3K–AKT signaling pathway, and viral protein interaction with cytokine (Figure 7C). For Erastin at 48 h, the DEPs were enriched in protein digestion and absorption at 48 h after Erastin treatment (Figure 7D). Compared to the SP600125 group, the DEPs were mainly enriched in ECM–receptor interaction, ferroptosis, and the FoxO signaling pathway at 24 h after Erastin treatment (Figure 7E) and were mainly enriched in oxidative phosphorylation at 48 h after Erastin treatment (Figure 7F). Taken together, the TNF and IL-17 signaling pathways and ECM–receptor interactions are more important in SP600125 treatment, whereas ECM–receptor interactions and the PI3K–AKT signaling pathway are more important in Erastin treatment. The details of proteins assessed in the GO enrichment and KEGG enrichment analyses are provided in Appendix A.

We performed KEGG enrichment analyses on up- or down-regulated DEPs, respectively. Compared to the NC group, the upregulated DEPs were mainly enriched in cytokine–cytokine receptor interaction, the IL-17 signaling pathway, the TNF receptor signaling pathway, the ECM–receptor interaction, and the PI3K–AKT signaling pathway at 24 h after SP600125 treatment (Appendix A); whereas the downregulated DEPs were mainly enriched in the mineral absorption at 24 h after SP600125 treatment (Appendix A). Compared to the NC group, the upregulated DEPs were mainly enriched in cytokine–cytokine receptor interaction, ferroptosis, viral protein interaction with cytokine, and the cysteine and methionine metabolism at 24 h after Erastin treatment (Appendix A); whereas the downregulated DEPs were mainly enriched in cytokine–cytokine receptor interaction, the PI3K–AKT signaling pathway, and the HIF signaling pathway at 24 h after Erastin treatment (Appendix A). Compared to the SP600125 group, the results indicated that the upregulated DEPs were mainly enriched in ferroptosis, lysosome, and the PPAR signaling pathway at 24 h after Erastin treatment (Appendix A); however, the downregulated DEPs were mainly enriched in viral protein interaction with cytokine, cytokine–cytokine receptor interaction, the HIF signaling pathway, the ECM–receptor interaction, and the PI3K–AKT signaling pathway at 24 h after Erastin treatment (Appendix A). Compared to the NC group, the upregulated DEPs were mainly enriched in the PI3K–AKT signaling pathway and steroid hormone biosynthesis at 48 h after SP600125 treatment (Appendix A), whereas the downregulated DEPs were mainly enriched in the ECM–receptor interaction and the PI3K–AKT signaling pathway at 48 h after SP600125 treatment (Appendix A). Compared to the NC group, the upregulated DEPs were mainly enriched in ferroptosis, the PPAR signaling pathway, and the cAMP signaling pathway at 48 h after Erastin treatment (Appendix A); whereas the downregulated DEPs were mainly enriched in the ECM–receptor interaction at 48 h after Erastin treatment (Appendix A). Compared to the SP600125 group, the results indicated that the upregulated DEPs were mainly enriched in ECM–receptor interaction and ferroptosis at 48 h after Erastin treatment (Appendix A); however, the downregulated DEPs were mainly enriched in oxidative phosphorylation at 48 h after Erastin treatment (Appendix A). Taken together, we found that upregulated DEPs were also involved in Erastin-induced ferroptosis but not downregulated DEPs.

To identify potential trends among the three groups, we performed cluster analysis on all DEPs, which were divided into five subclusters based on protein expression (Figure 8A). As the effects of SP600125 and Erastin on ferroptosis are opposite, we focused on proteins that exhibited different expression patterns between the SP600125 and Erastin groups. We found that subcluster_1 and subcluster_3 fit this expression pattern (Figure 8B). GO enrichment analysis of subcluster_1 indicated that the DEPs were mainly enriched in carboxylic acid metabolic process, oxoacid metabolic process, organic acid metabolic process, amino acid metabolic process, and oxidoreductase activity (Figure 8C), while KEGG enrichment analysis of subcluster_1 showed that the DEPs were mainly enriched in aminoacyl-tRNA biosynthesis, ferroptosis, biosynthesis of cofactors, and protein processing in the endoplasmic reticulum (Figure 8D). GO enrichment analysis of subcluster_3 revealed that the DEPs were mainly enriched in the immune system process, small molecule metabolic process, carboxylic acid metabolic process, oxoacid metabolic process, and organic acid metabolic process (Figure 8E), while KEGG enrichment analysis of subcluster_3 indicated that the DEPs were enriched in the ECM–receptor interaction, ferroptosis, and cell adhesion molecules (Figure 8F).

### 3.7. Conjoint Analysis of DEGs and DEPs and Protein–Protein Interaction Network

Conjoint analysis can further validate DEGs and DEPs, addressing the limitations of relying solely on single omics analyses. In this study, we identified 316 DEGs and DEPs associated with the SP600125 vs. NC group comparison at 24 h after SP600125 treatment, of which 1 DEG (1 up-regulated and 0 down-regulated) exhibited consistent expression with its corresponding DEPs (Figure 9A). Similarly, we found 294 DEGs and DEPs at 48 h after SP600125 treatment, with 0 DEGs exhibiting consistent expression with their corresponding DEPs (Appendix A).

Furthermore, we found that 674 DEGs and DEPs were associated with the Erastin vs. NC group comparison at 24 h after Erastin treatment, with 20 DEGs (11 up-regulated and 9 down-regulated) showing consistent expression with their corresponding DEPs (Figure 9B). At 48 h after Erastin treatment, we found 654 DEGs and DEPs, with 0 DEGs exhibiting consistent expression with their corresponding DEPs (Appendix A).

We observed 623 DEGs and DEPs associated with the Erastin vs. SP600125 group comparison at 24 h after treatment, of which 21 DEGs (10 up-regulated and 11 down-regulated) exhibited consistent expression with their corresponding DEPs (Figure 9C). At 48 h after treatment, we found 654 DEGs and DEPs, with 0 DEGs exhibiting consistent expression with their corresponding DEPs (Appendix A).

Finally, we conducted STRING analysis to examine the potential interaction network of all 31 named DEGs and DEPs associated with the three groups. The analysis revealed a protein–protein interaction (PPI) network consisting of 31 DEGs and DEPs (Figure 9D). Notably, we found *HMOX1* and *CXCL8* were positioned in the network, which were interacting with ferroptosis-related genes GCLM or F2R, respectively, which suggests *HMOX1* and *CXCL8* may be potential candidate genes involved in the regulation of ferroptosis and JEV proliferation (Figure 9D).

To further explore the relationships between the groups, we performed GO and KEGG analysis on the DEGs and DEPs derived from the association analysis between the groups. The results revealed significant GO enrichment in processes such as protein targeting to vacuole, intracellular sterol transport, golgi subcompartment, and lipid transport between the SP600125 and NC groups at 24 h after SP600125 treatment (Figure 10A), and we found several significantly enriched KEGG pathways: inositol phosphate metabolism, peroxisome, proteasome, fatty acid degradation, synaptic vesicle cycle, and cytokine–cytokine receptor interaction (Figure 10B). Compared to the NC group, the results indicated that the immunological synapse, oxidoreductase activity, serine family amino acid biosynthetic process, and metabolic process were significantly enriched between the Erastin and NC groups at 24 h after Erastin treatment (Figure 10C), and the enriched KEGG pathways included the HIF-1 signaling pathway, complement and coagulation cascades, starch and sucrose metabolism, ribosome biogenesis in eukaryotes, terpenoid backbone biosynthesis, glycine, serine, and threonine metabolism, and proteasome (Figure 10D). Compared to the SP600125 group, the results indicated that the pteridine-containing compound metabolic process, and serine family amino acid biosynthetic process, Arp2/3 complex-mediated actin nucleation were significantly enriched between the Erastin and SP600125 groups at 24 h after reagent treatment (Figure 10E). The significantly enriched KEGG pathways included the HIF-1 signaling pathway, mineral absorption, and complement and coagulation cascades (Figure 10F).

## 4. Discussion

Pigs are naturally infected by mosquitoes carrying JEV. The virus first propagates in skin epithelial cells and lymph nodes and then infects peripheral organs such as the kidney, liver, and spleen, causing transient viremia. Following this, the neurotropic virus spreads to the central nervous system. Viruses can induce ferroptosis in host cells through various mechanisms in order to promote their replication and survival. In the current study, we observed that mRNA levels of the anti-ferroptosis-related genes *GPX4*, *NFE2L2*, *NQO1*, *GCLM,* and *GCLC* significantly decreased following JEV infection. Conversely, the expression of *PTGS2* significantly increased during JEV infection. These findings suggest that JEV may induce ferroptosis through downregulating anti-ferroptosis-related genes and upregulating pro-ferroptosis-related genes. This dual modulation could serve as a mechanism through which JEV promotes its replication and survival within host cells. LDH is often used as a biomarker for cell damage or death, and LDH activity can be influenced by the level of ROS in the cell. LDH also serves as an important biomarker for ferroptosis [32,33]. MDA is another widely used marker for ferroptosis, as it is a byproduct of lipid peroxidation; during ferroptosis, the accumulation of lipid peroxides leads to the decomposition of these peroxides into toxic derivatives such as MDA [34,35,36]. In the present study, we found that the levels of MDA, ROS, and LDH were significantly increased after JEV infection. These results indicated that JEV can induce ferroptosis in PK-15 cells. RSL3 is an agonist for ferroptosis that acts by suppressing GPX4 [37,38]. Erastin is also an agonist for ferroptosis that acts by promoting ROS accumulation [39,40,41]. In this study, we found that the effects of Erastin were better than RSL3 in terms of inhibiting JEV proliferation. SP600125 is an antagonist for ferroptosis and can also suppress the JNK pathway, while Ferrostatin-1 inhibits the ferroptosis by protecting the membrane lipid peroxidation [42,43,44]. The JNK pathway is also a key pathway for JEV proliferation. A previous study demonstrated that knocking down JNK or using low doses of SP600125 can inhibit the proliferation of JEV [45]. However, in our study, we observed that SP600125 at high doses (10 μM and 20 μM) promotes JEV proliferation, whereas a low dose (1 μM) does not. Based on these findings, we consider that at low doses, SP600125 suppresses JNK activity without affecting ferroptosis. Conversely, at high doses, SP600125 inhibits both ferroptosis and JNK. The opposing effects of the two pathways on JEV proliferation suggest that the influence of ferroptosis may overshadow the impact of JNK at high SP600125 concentrations, ultimately promoting JEV proliferation. Ferrostain-1 is an established ferroptosis inhibitor; the effect of Ferrostain-1 can also demonstrate that the inhibition of ferroptosis can promote JEV proliferation. In the present study, we found that the effects of SP600125 were higher than those of Ferrostatin-1 in terms of promoting JEV proliferation; therefore, we chose SP600125 for the subsequent experiments. These results imply that pretreatment of PK-15 with Erastin promotes ferroptosis via ROS accumulation, allowing JEV to be controlled effectively. Iron is an essential element for supporting basic cellular processes and is crucial to support the growth, virulence, and pathogenicity of viruses [46]. Iron usually acts as an essential element for oxygen transfer and serves as both an electron donor and acceptor [47]. Overall, iron is abundantly present in mammals, either existing freely or bound to hepcidin. It exhibits the ability to modulate the replication of various viral infections across different organisms.

The results of the RNA-seq and 4D-DIA proteomics analyses revealed significant enrichment of the TNF signaling pathway, cytokine–cytokine receptor interaction, the IL-17 signaling pathway, the Toll-like receptor signaling pathway, viral protein interaction with cytokine, the PI3K–AKT signaling pathway, ferroptosis, ECM–receptor interaction, and the chemokine signaling pathway after treatment with the ferroptosis antagonist SP600125 or agonist Erastin. These pathways are likely implicated in the ferroptosis-induced regulation of JEV proliferation. At 24 h after JEV infection, viral protein interaction with cytokines, cytokine–cytokine receptor interaction, the IL-17 signaling pathway, and the Toll-like receptor signaling pathway were presented in both SP600125 and Erastin treatment. In the SP600125 treatment, the genes *CCL2*, *CCL5*, *CXCL10*, *CXCL11*, and *CCR7* were involved in viral protein interaction with cytokine, whereas in the Erastin treatment, the genes *CCL3L1*, *CXCL8*, *CCL5*, *IL22RA1*, *CCL4*, *CXCL14*, *IL6*, *ACKR3*, *CCL25*, *TNF*, and *GHSR* were involved in viral protein interaction with cytokine. *CCL5* is the gene presented in both the Erastin and SP600125 treatment groups, and the gene *CCL5* is up-regulated in both groups compared to the NC group. In the SP600125 treatment, the genes *CCL5*, *IL21R*, *CCL2*, *TNFSF15*, *CXCL10*, *CXCL11*, *IL27*, *IL28B*, and *CCR7* were involved in cytokine-cytokine receptor interaction; in the Erastin treatment, the genes *IL6*, *CCL3L1, OSMR*, *CCL5*, *IL13RA1*, *CCL4*, *CXCL14*, *IL22RA1*, *IFNAR1*, *ACKR3*, *CCL25*, *TNF*, *GHSR*, *INHBE*, *GDF15*, *TNFSF13B*, *IL23A*, *IL28B*, and *CXCL8* were involved in cytokine–cytokine receptor interaction. In the SP600125 treatment, the genes *CCL2*, *CXCL10*, *FOS*, *FOSB*, and *S100A8* were involved in the IL-17 signaling pathway; in the Erastin treatment, the genes *IL6*, *CXCL8*, *S100A8*, *MAPK11*, *FOS*, *FOSB*, and *TNF* were involved in the IL-17 signaling pathway. *FOS* and *FOSB* are the genes presented in both the Erastin and SP600125 treatment groups, and the gene *CCL5* is downregulated in both groups compared to the NC group. In the SP600125 treatment, the genes *FOS*, *CCL5*, *CXCL10*, and *CXCL11* were involved in the Toll-like receptor signaling pathway, whereas in the Erastin treatment, the genes *IL6*, *CCL3L1*, *CXCL8*, *CCL5*, *CCL4*, *TLR1*, *MAPK11*, *IFNAR1*, *FOS*, *TNF*, and *CTSK* were involved in the Toll-like receptor signaling pathway. At 48 h after JEV infection, the TNF signaling pathway and IL-17 signaling pathways were presented in both SP600125 and Erastin treatment. In the SP600125 treatment, the genes *VCAM1*, *CSF2*, and *PTGS2* were involved in the TNF signaling pathway, whereas in the Erastin treatment, the genes *ENSSSCG00000061605* and *ENSSSCG00000016254* were involved in the TNF signaling pathway. In the SP600125 treatment, the genes *CSF2*, *PTGS2*, and *S100A8* were involved in the IL-17 signaling pathway; whereas in the Erastin treatment, the genes *ENSSSCG00000061605* and *ENSSSCG00000016254* were involved in the IL-17 signaling pathway (Appendix A). Taken together, we found most of the DEGs were not co-regulated in the pathways both presented in the SP600125 and Erastin groups. The TNF-pathway can trigger ferroptosis in cells under conditions of inflammatory and oxidative stress [48,49]. A previous study has reported that inhibiting microglia ferroptosis will decrease the release of TNF-α, alleviating downstream effects such as oligodendrocyte precursor cell necroptosis [50]. The TNF signaling pathway is associated with the upregulation of cytokines and chemokines, which are essential for recruiting immune cells to the site of infection. This response can lead to both protective and detrimental effects, influencing disease outcomes [51,52]. JEV infection activates TLR3 and TLR2, which play a role in initiating the TNF signaling cascade and can activate the NF-κB pathway; as a result, the inflammatory response and production of cytokines are enhanced [52,53]. The IL-17 signaling pathway plays a significant role in the regulation of ferroptosis. IL-17A is the main cytokine in the IL-17 signaling pathway, which can stimulate the production of IL-6. Elevated levels of IL-6 can promote oxidative stress and lipid peroxidation, thus leading to ferroptosis [54]. IL-17A can activate the TNF signaling pathway and further drive the inflammatory response, leading to increased ferroptosis [48,55]. JEV infection can activate the IL-17 signaling pathway in macrophages, leading to the upregulation of various inflammatory genes and cytokines [56]. ECM–receptor interactions can influence ferroptosis via modulating iron metabolism and cellular response to oxidative stress, and ECM detachment can also confer resistance to ferroptosis via alterations in iron uptake and storage [48,57]. The PI3K–AKT signaling pathway can be activated during PRV infection through the engagement of receptor tyrosine kinases (RTKs) and is involved in controlling cell apoptosis [58]. The FeHV-1 can modify the PI3K/Akt/mTOR axis, as well as its function in crucial physiological processes like autophagy, apoptosis, or the IFN induction cascade [59]. AKT can also be activated by TLR2 and promotes the expression of GPX4, protecting cells from lipid peroxidation. Inhibition of the PI3K–AKT pathway can decrease the GPX4 levels, sensitizing cells to ferroptosis [60,61]. The PI3K–AKT pathway can also activate the NRF2, elevating ferroptosis resistance [62,63,64]. Toll-like receptors (TLRs), particularly TLR2, also play a critical role in mediating the inflammatory response to JEV. The TLR2–PI3K–AKT signaling axis has been identified as a key pathway that regulates the inflammatory response in microglia during JEV infection [53]. Chemokines are a family of small cytokines that play a crucial role in regulating inflammation and immune responses. Recent studies have also implicated chemokine signaling pathways in the regulation of ferroptosis. Chemokines such as CXCL8 and CCL2 are upregulated during inflammation and have been shown to sensitize cells to ferroptosis [65]. Chemokines also play important roles in controlling JEV, as chemokines such as CCL2, CCL3, CCL4, and CXCL1 are involved in the recruitment of immune cells to the (central nervous system) CNS during JEV infection. Their expression is dynamically regulated and correlated with the severity of the inflammatory response and disease progression. The chemokine receptor CCR5 has been shown to modulate the CNS infiltration of regulatory T cells, influencing the outcome of JEV infection. CCR5 deficiency can exacerbate the disease, indicating its protective role in regulating immune responses [66].

JEV infection can trigger the inflammatory response through the activation of immune cells and the release of various cytokines. This response is essential for controlling the virus but can also lead to tissue damage. JEV infection can stimulate various pro-inflammatory cytokines, such as IL-1β, IL-6, TNF-α, and type-I interferons. These cytokines can mediate inflammation and the recruitment of immune cells to the infection site [67,68]. The inflammatory response and cytokine production play pivotal roles in the pathogenesis of JEV infection. While these responses are essential for controlling the virus, they can also lead to significant tissue damage and contribute to the severity of the disease. Understanding the balance between protective and pathological inflammatory responses is crucial in developing effective treatments for JEV and related viral infections.

Notably, two subclusters with opposite expression patterns emerged from the RNA-seq cluster analysis. Of the RNA-seq clusters, genes in Subcluster_1 were upregulated by Erastin, while genes in Subcluster_2 were downregulated by Erastin. These results indicated that Erastin induced ferroptosis in PK-15 cells and suppressed JEV proliferation by activating the Toll-like receptor signaling pathway, the P53 signaling pathway, viral protein interaction with cytokines, the IL-17 signaling pathway, and activating the TNF signaling pathway, as a result enhancing the inflammatory response and cytokine production and inhibiting the proliferation of JEV. Among these pathways, the Toll-like receptor signaling pathway and P53 signaling pathway were presented in both subclusters. In Subcluster_1, the genes IL6, *CCL3L1*, *CXCL8*, *CCL5*, *CCL4*, *MAPK11*, and *TNF* were involved in the Toll-like receptor signaling pathway, whereas in Subcluster_2, the genes *TLR1*, *CTSK*, *IFNAR1*, and *FOS* were involved in the Toll-like receptor signaling pathway. In Subcluster_1, the genes *SIVA1*, *SESN2*, *BBC3*, *GADD45A*, and *PMAIP1* were involved in the P53 signaling pathway, whereas in Subcluster_2, the genes *SESN3*, *CCNG2*, and *SERPINE1* were involved in the P53 signaling pathway. However, no genes of the two subclusters were repetitive in the Toll-like receptor signaling pathway and P53 signaling pathway (Appendix A). When we analyzed all consistent expression DEGs and DEPs for PPI, we identified HMOX1 as a potential candidate gene that interacts with CBS and GCLM (Figure 8E) and identified CXCL8 as a potential candidate gene that interacts with F2R. These results suggest that GCLM and CXCL8 may play crucial roles in ferroptosis. HMOX1 overexpression can increase sensitivity to ferroptosis agonists like Erastin in various cell types [69,70,71]; furthermore, HMOX1 catalyzes the degradation of heme, releasing ferrous iron, which can promote ferroptosis via the Fenton reaction and lipid peroxidation [35,70,71]. GCLM is a key antioxidant that protects cells from oxidative stress and ferroptosis. GLCM works in conjunction with GLC to catalyze the first step in GSH synthesis, where the availability of GSH is critical for cellular defense against ferroptosis [17,72]. In this study, we found that GLCM is upregulated by the ferroptosis agonist Erastin; this result is similar to that in a previous study, where it was reported that ferroptosis accompanies the induction of genes that can restrict the execution of ferroptosis [73]. CBS plays a dual role in ferroptosis, acting as a protective factor against ferroptosis through its involvement in the transsulfuration pathway and GSH synthesis [74,75]. CXCL8 can interact with F2R, which is a G protein-coupled receptor that can regulate iron metabolism and affect the ferroptosis. F2R activation can also lead to the generation of ROS, which is pivotal in the ferroptosis pathway [76,77].

Ferroptosis, as a form of regulated cell death characterized by iron accumulation of lipid peroxidation, has a complex role in viral infections. Previous studies have shown that Coxsackievirus B3 (CVB3) and Influenza A Virus (IAV) can utilize TFR1 to facilitate their entry into host cells, leading to iron uptake and promoting ferroptosis due to the elevated levels of ROS [30]. Viruses can also induce iron overload in host cells, resulting in increased intracellular iron levels that trigger ferroptosis. SARS-CoV-2, ZIKV, HSV, and CV-A6 infections can downregulate the expression of GPX4, leading to lipid peroxides and facilitating viral replication [11,12]. In this study, we found that JEV can also promote ferroptosis in PK-15 cells. The ferroptosis agonist Erastin has been found to inhibit PEDV in Vero cells, and treatment with Erastin during the viral replication stage significantly reduced PEDV levels [78]. However, in colorectal cancer cells, Erastin was shown to induce ferroptosis in a dose- and time-dependent manner, as treatment with 20 μM Erastin for 48 h elevated iron levels and lipid peroxidation [79]. In the present study, the results indicated that Erastin treatment can significantly inhibit the proliferation of JEV.

## 5. Conclusions

Our results indicated that JEV infection promotes ferroptosis via suppressing the expression of the anti-ferroptosis-related genes *NFE2L2*, *GCLC*, *GCLM*, *NQO1*, and *GPX4* while also inducing the expression of the ferroptosis-related gene *PTGS2* and the levels of LDH, ROS, and MDA. Treatment with the ferroptosis agonist Erastin treatment significantly suppressed the proliferation of JEV, while the ferroptosis antagonist SP600125 promoted JEV proliferation. Erastin treatment can suppress JEV via the ECM–receptor interaction, cytokine–cytokine receptor interaction, IL-17, TNF, Toll-like receptor signaling pathways, etc. In summary, treatment with the ferroptosis agonist Erastin leads to increased oxidative stress and lipid peroxidation within the infected cells, creating an environment that is detrimental to viral survival and replication. The inhibitory effects of Erastin are most pronounced during the replication phase, suggesting that it disrupts the viral life cycle effectively.

## Figures and Tables

**Figure 1 animals-14-03516-f001:**
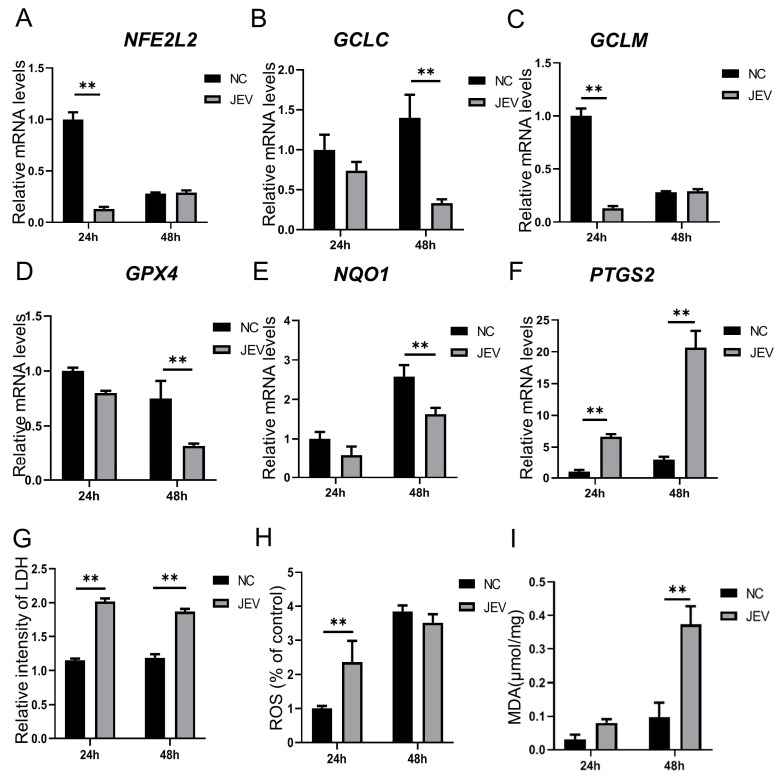
Ferroptosis-related mRNA and indicator levels induced by JEV infection: (**A**–**F**) mRNA levels of ferroptosis-related genes, *n* = 3; (**G**) LDH levels induced by JEV; (**H**) ROS levels induced by JEV; and (**I**) MDA levels induced by JEV, *n* = 8. ** *p* < 0.01.

**Figure 2 animals-14-03516-f002:**
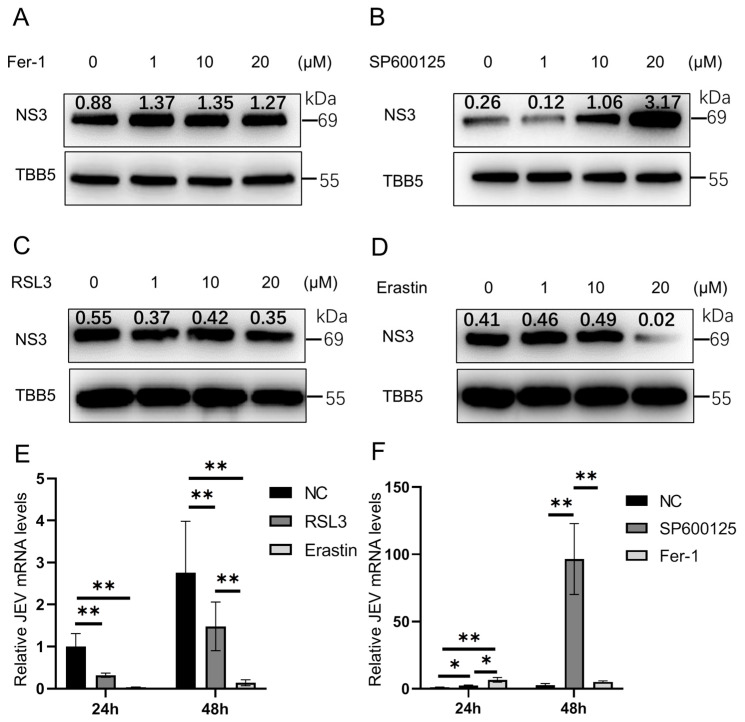
JEV NS3 protein levels and JEV mRNA levels regulated by ferroptosis agonists or antagonists: (**A**) protein levels of NS3 treated with ferroptosis antagonist Fer-1; (**B**) protein levels of NS3 treated with ferroptosis antagonist SP600125; (**C**) protein levels of NS3 treated with ferroptosis agonist RSL3; (**D**) protein levels of NS3 treated with ferroptosis agonist Erastin; (**E**) JEV mRNA levels treated with ferroptosis agonist RSL3 (20 μM) and Erastin (20 μM), *n* = 3; and (**F**) JEV mRNA levels treated with ferroptosis antagonist SP600125 (20 μM) and Fer-1 (20 μM), *n* = 3. * *p* < 0.05, ** *p* < 0.01.

**Figure 3 animals-14-03516-f003:**
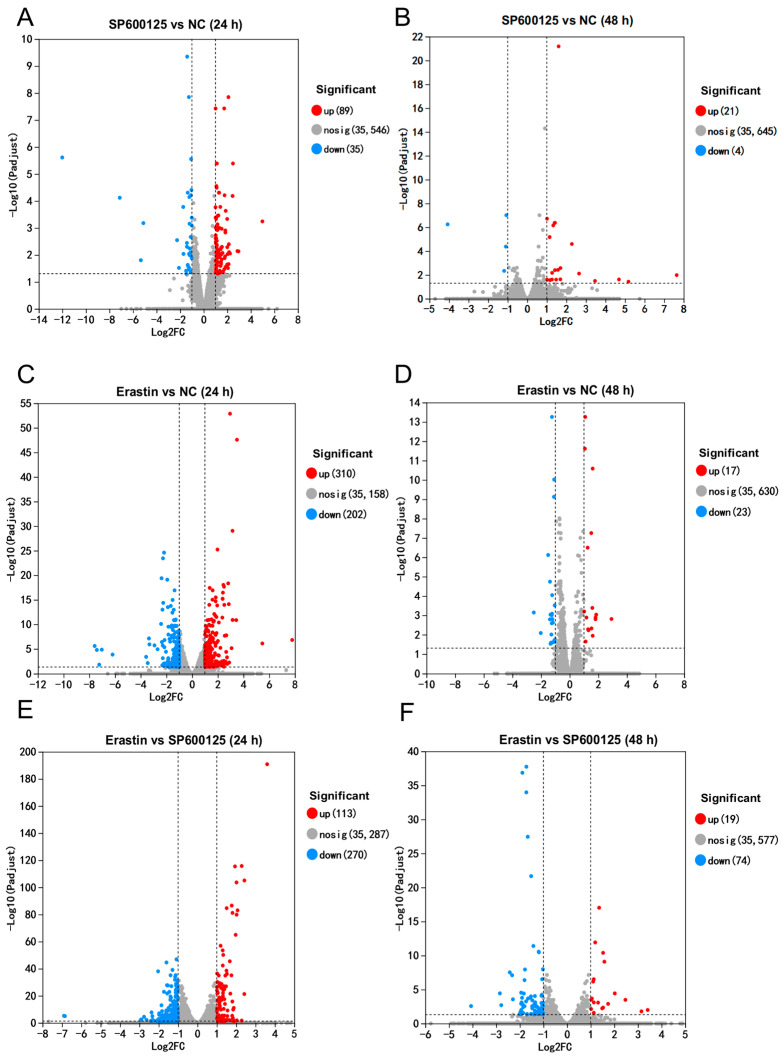
Volcano plot of DEGs between different groups: (**A**,**B**) DEGs between SP600125 and NC groups. (**C**,**D**) DEGs between Erastin and NC groups; and (**E**,**F**) DEGs between Erastin and SP600125 groups.

**Figure 4 animals-14-03516-f004:**
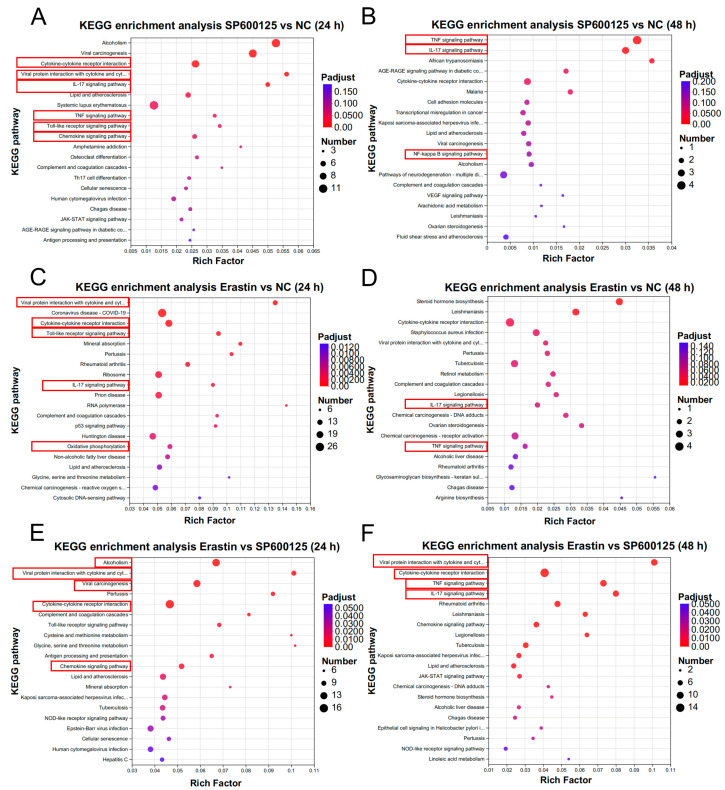
Pathways enrichment analysis of DEGs: (**A**,**B**) The top 20 KEGG enrichment pathways between SP600125 and NC groups; (**C**,**D**) The top 20 KEGG enrichment pathways between Erastin and NC groups; and (**E**,**F**) The top 20 KEGG enrichment pathways between Erastin and SP600125 groups.

**Figure 5 animals-14-03516-f005:**
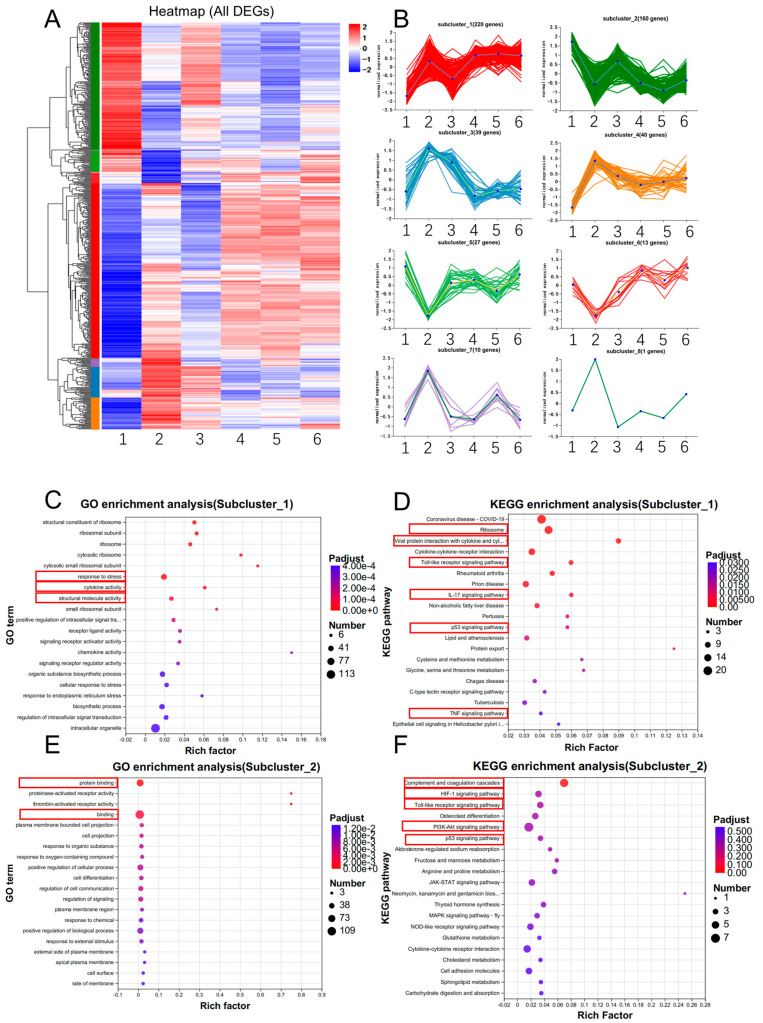
DEGs expression pattern for JEV-infected PK-15 cells treated with ferroptosis agonist or antagonist: (**A**) Heatmap of DEGs, *X*-axis: 1 = NC 24 h, 2 = Erastin 24h 3 = SP600125 24 h, 4 = NC 48 h, 5 = Erastin 48 h. 6 = SP600125 48 h; (**B**) eight subclusters of cluster analysis. *Y*-axis: log10 (expression); (**C**) GO functional enrichment analysis of DEGs in subcluster_1; (**D**) KEGG enrichment analysis of DEGs in subcluster_1; (**E**) GO functional enrichment analysis of DEGs in subcluster_2; and (**F**) KEGG enrichment analysis of DEGs in subcluster_2.

**Figure 6 animals-14-03516-f006:**
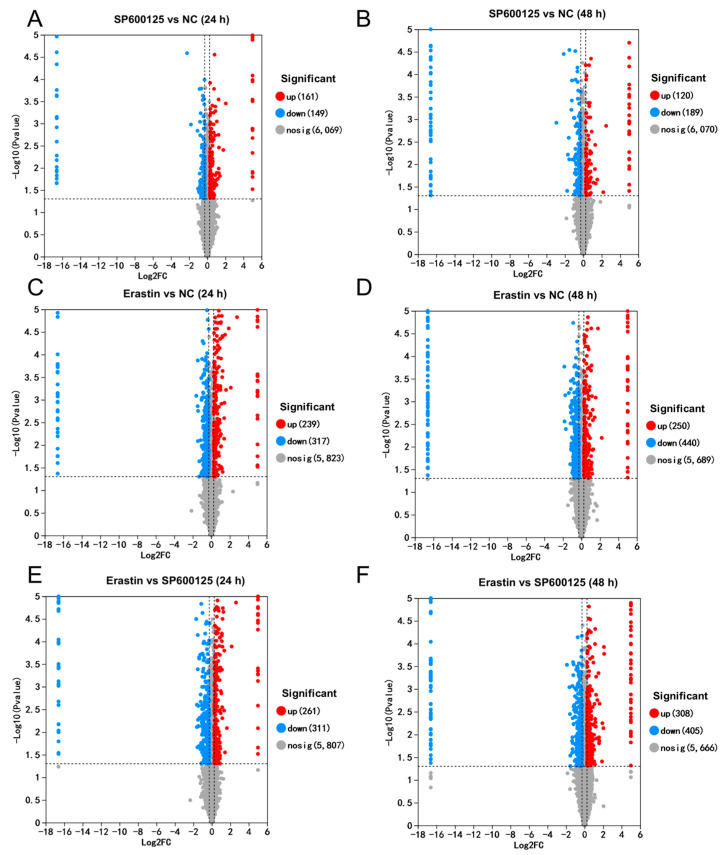
Volcano plot of DEPs between different groups: (**A**,**B**) DEPs between SP600125 and NC groups; (**C**,**D**) DEPs between Erastin and NC groups; and (**E**,**F**) DEPs between Erastin and SP600125 groups.

**Figure 7 animals-14-03516-f007:**
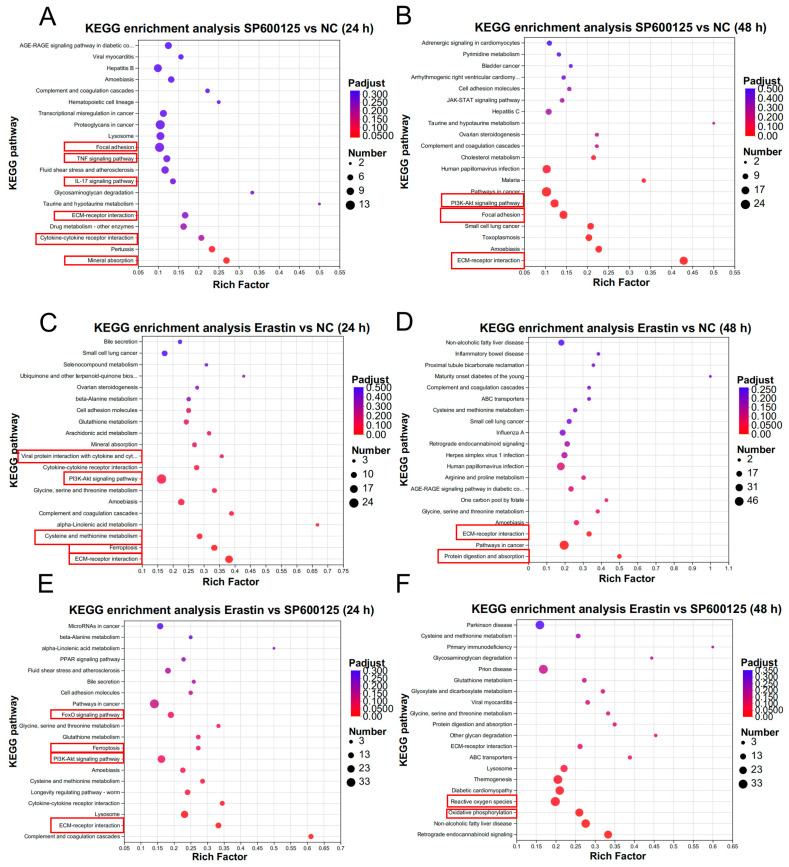
Pathways enrichment analysis results for DEPs: (**A**,**B**) The top 20 KEGG enrichment pathways between SP600125 and NC groups; (**C**,**D**) The top 20 KEGG enrichment pathways between Erastin and NC groups; and (**E**,**F**) The top 20 KEGG enrichment pathways between Erastin and SP600125 groups.

**Figure 8 animals-14-03516-f008:**
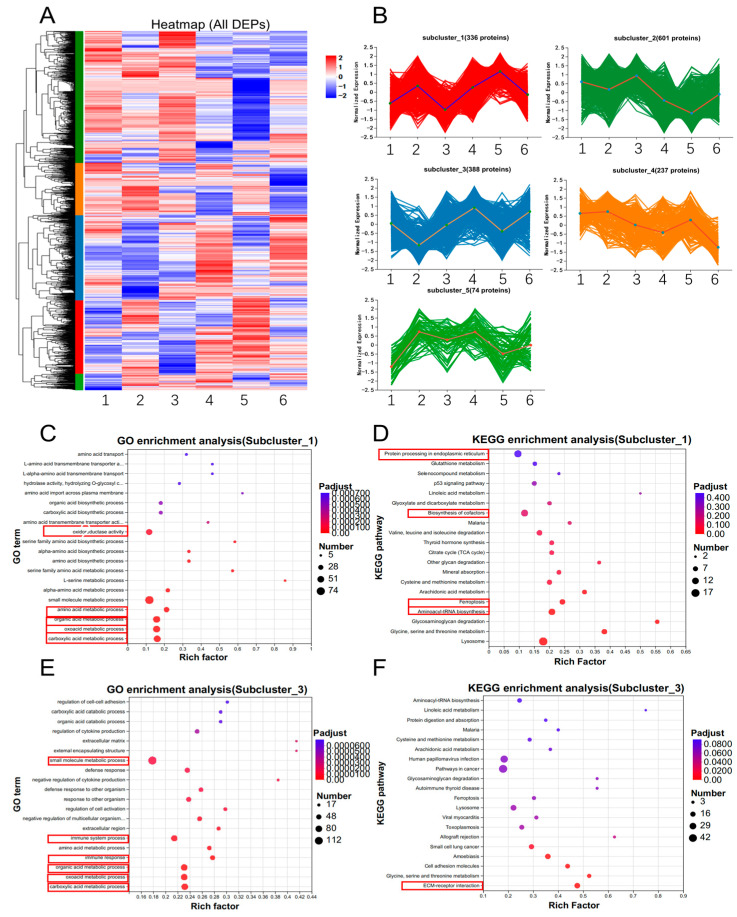
DEPs expression patterns for JEV-infected PK-15 cells treated with ferroptosis agonist or antagonist; (**A**) heatmap of DEPs, *X*-axis: 1 = NC 24 h, 2 = Erastin 24 h 3 = SP600125 24 h, 4 = NC 48 h, 5 = Erastin 48 h. 6 = SP600125 48 h; (**B**) five subclusters of cluster analysis. *Y*-axis: log10 (expression). (**C**) GO functional enrichment analysis of DEPs in subcluster_1; (**D**) KEGG enrichment analysis of DEPs in subcluster_1; (**E**) GO functional enrichment analysis of DEPs in subcluster_3; and (**F**) KEGG enrichment analysis of DEPs in subcluster_3.

**Figure 9 animals-14-03516-f009:**
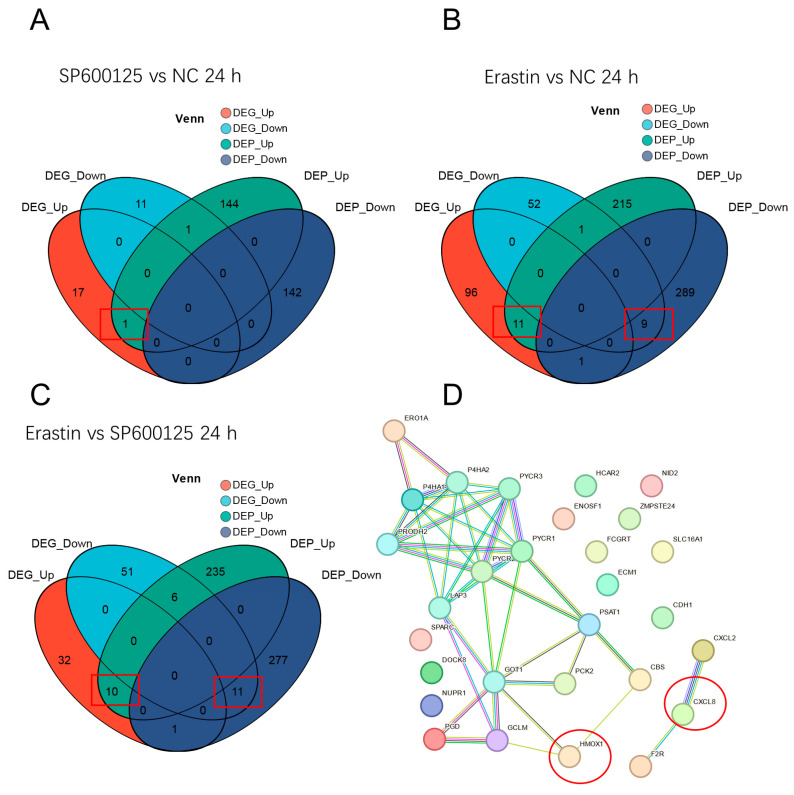
Venn diagram of DEGs and DEPs and protein–protein interaction network: (**A**) venn diagram of DEGs and DEPs between SP600125 and NC groups (**B**) venn diagram of DEGs and DEPs between Erastin and NC groups; (**C**) venn of DEGs and DEPs between Erastin and SP600125 groups. The top 20 enrichment pathways between Erastin and NC groups and (**D**) protein–protein interaction network analysis. Each node represents a gene, and the thickness of the line connecting two nodes indicates the strength of the protein interaction.

**Figure 10 animals-14-03516-f010:**
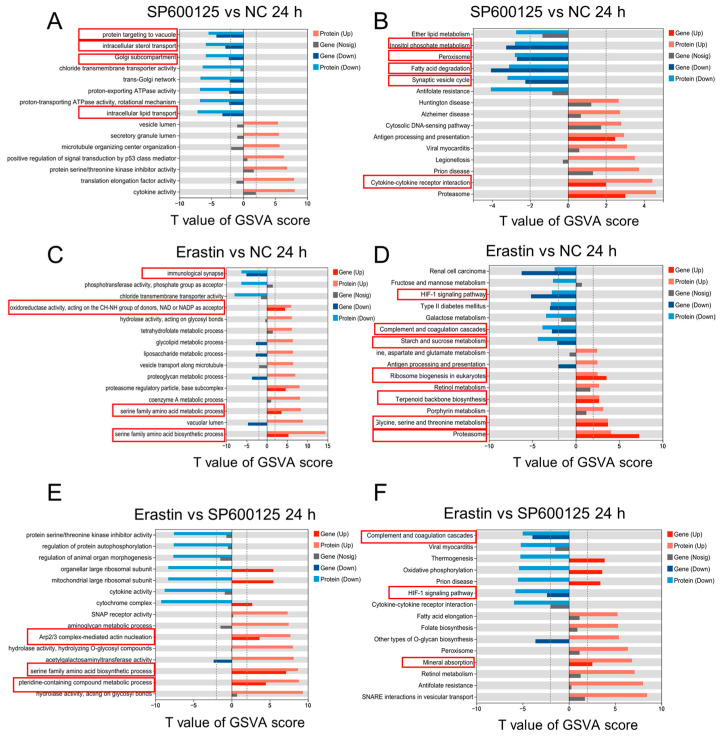
GO and KEGG Pathways enrichment analyses of DEGs and DEPs: (**A**) the top 15 GO functional enrichment analyses between the SP600125 and NC groups; (**B**) the top 15 KEGG enrichment pathways between the SP600125 and NC groups; (**C**) the top 15 GO functional enrichment analyses between the Erastin and NC groups; (**D**) the top 15 KEGG enrichment pathways between the Erastin and NC groups; (**E**) the top 15 GO functional enrichment analyses between the Erastin and SP600125 groups; and (**F**) the top 15 KEGG enrichment pathways between the Erastin and SP600125 groups.

## Data Availability

Data are available from the corresponding author upon reasonable request.

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
