# Peer review of "Transcriptome and Proteome Analyses Revealed Differences in JEV-Infected PK-15 Cells in Response to Ferroptosis Agonists and Antagonists"

_animals, 2024, doi:10.3390/ani14233516_

Round 1
Reviewer 1 Report
Comments and Suggestions for Authors
Thank you very much for allowing me to review your manuscript. I found it to be very interesting and potentially important, but I do have a number of questions and comments that I'd like you to address:
1. First, your title is very vague. Instead of stating that the transcriptome and proteomic [analysis] revealed the function of ferroptosis, it would be better to state what that function is. Second, I'm not sure what your paper did. What I understand that you paper did was to identify genes and proteins related to ferroptosis that were differentially expressed in the presence or absence of ferroptosis agonists and antagonists. Perhaps stating that would be more clear and correct in the title.
2. In line 25-26, in the abstract, you state that "RNA-seq analysis identified 806 differentially expressed genes and 951 differentially expressed proteins in PK-15 cells 24 h after treatment with the ferroptosis agonist and inhibitor, respectively". I'm confused by this sentence. What does the 806 and 951 represent? The number of genes and proteins that were differentially expressed, respectively, when treated with either the agonist or the inhibitor, or the number of genes and proteins when treated with the agonist and inhibitor, respectively?
3. Could you please add references to your statements in the first paragraph of the introduction?
4. On line 80, you state "After incubating the cells for 2 hrs.." Which cells do you mean, the PK-15 or the BHK-21 cells? Also, what do you mean by MOI=1?
5. It's unclear on line 81 how the PK-15 cells were infected. Please clarify.
6. In section 2.2, you describe how you used RT-qPCR for 6 specific genes. However, I only understood why you chose these specific genes when I got to page 16 in the Discussion. Please move the first paragraph of page 16 into the introduction so the reader can understand in the methods why you chose these specific genes.
7. On line 92, what does the BCA method mean? Can you briefly describe?
8. On line 117, what is the RQN value and why is it important?
9. In your statistical analysis (section 2.2) you state that p<0.05 was considered statistically significant. Is this only for the RT-qPCR data? Did you use any correction for multiple testing, perhaps internally? Figure 4, 5,7, 8 all feature a legend that says "Padjust" that makes me think that the pathway analyses must have adjusted the p-values for multiple testing. Can you confirm that, and state what method they used?
10. On line 140, you state that you used One-Way ANOVA for your statistical comparison. I suppose this is for the RT-qPCR analysis in Figure 1. However, even there, I would suspect that you used a two-way ANOVA, because there are two factors: infection status and time (24 h vs 48 h).
11. I was surprised to see that the RT-qPCR analysis was only conducted to compare gene expression for the samples with or without JEV infection, but not in the presence of the agonist or antagonist. Why is that? I expected to see the results in Figure 1 in the case of treatments with both agonist and antagonist.
12. In your RNA sequencing analysis, you compare the differential gene and protein expression in the Negative control group to the same with the treatment with the agonist and the antagonist. Is the Negative Control the same as in Figure 1 where the cells have not been infected with JEV? If that is correct, then all the differences you see between NC and the treatment groups are a combination of changes due to JEV infection and the treatment, and the two are confounded, so it's not possible to separate the two. Did you perhaps mean as Negative Control group cells that have been infected with JEV, but not treated with either the agonist or the antagonist? That would make more sense, and then the results you show are correctly showing the effect of the agonist and the antagonist. However, then you shouldn't call it Negative Control, but call it just JEV. Which one of these is the correct interpretation? I hope that it is the later, since you state on line 198 that JEV-infected PK-15 cells begin to die at 48 hours. I have a similar question about the proteomics analysis. Please clarify!
13. In Figure 3, and page 6 and 7 you talk about the many differentially expressed genes at 24 and 48 hrs in the different groups. Can you clarify what proportion of the genes differentially expressed at 48 hr were a subset of the genes differentially expressed at 24 hr, so consistent between the timepoints? I have a similar question about Figure 6.
14. On line 214, you introduce the EggNOG classification. Can you explain what that classification is?
15. In section 3.4, you look at the Gene Expression Pattern analysis, you seem to be combining the results from 24 hrs and 48 hrs in each of the treatments. Is that justified? Are they consistent enough to do that?
16. In Figure 4, you show the results of the pathway enrichment analysis of the DEGs. However, could you also show the direction of the change, whether the genes in an enriched pathway are down- or up-regulated, either by showing the average log-fold change for the DEGs in each pathway, or by using the enrichment plot in GSEA? Otherwise, it's hard to tell what the direction of the impact is on gene expression! Perhaps you could do what you did on Figure 10, but of course just for DEGs. The same would apply for Figure 7 just for DEPs.
17. On line 249, you state that you performed cluster analysis. How was this performed, with what software? These should really go into the Methods section on statistical analysis. Why did you choose subcluster_1 and subcluster_2 for further analysis? The same applies to Figure 8 and line 355.
18. On line 303, I'm not sure where the numbers 425 and 565 come from. Are these the genes that are specific DEPs to the specific treatment? Please clarify!
19. On lines 305 and 307, you mention DEGs again. Do you mean DEPs here?
20. In the last paragraph of page 11, please discuss your results in the order of the subplots of Figure 7, so you don't have to jump around Figure 7A, then 7C, then 7E, then 7B, 7D and 7F.
21. I'm unclear on where the numbers are coming from in the first and second paragraphs of section 3.6 in terms of the numbers of DEGs and DEPs identified. They don't seem to match up with previous figures in an intuitive way.
22. On line 394, you stat that "31 named DEGs and DEPs associated with the three groups. However, when I add up all the genes that have consistent expression between DEGs and DEPs in each of the three group pairings, that is 1+11+9+10+11=41 genes. Why do you state 31 instead of 41? What am I missing?
23. On line 396, you state that the genes "HMOX1 and CXCL8 was positioned at crucial nodes in the network". How was that determined? They certainly don't have the highest connectivity. What metric was used to assess how crucial each node is in the network? That network analysis should also be explained in the methods under statistical analysis.
24. On Figure 10, you show the conjoint pathway enrichment analysis of DEGs and DEPs. This one doesn't have any of the pathways highlighted. How do you know which ones to focus on, and mention in the text?
25. Again, lines 438-478 should go into the Introduction.
26. On lines 493-497, you list a whole bunch of pathways that were enriched "after ferroptosis inhibitor SP600125 or Erastin treated". So it this just a list of all the pathway that were enriched in one or the other treatment? Wouldn't it be important to also see which were enriched in both treatments, and how the two treatments contrasted in terms of which pathways were enriched, and also whether those pathways were up- or down-regulated? I don't presently see this comparison. The same would apply to the list of pathways on lines 550-551. Are these the pathways represented by both subclusters? How do they change in response to the different treatments?
27. Finally, can you speculate on whether Erastin, which promotes ferroptosis, could be used to treat JEV, or not really, because it would to cell death that would be worse than the infection?
28. In Figure S1 and S3, what are the EggNOG categories? What do they represent?
29. In Figure S2, in subplot B, it looks like all the adjusted p-values are not significant. Why are all the Rich factor values 1 for the last 7 pathways? Also, in subplot D, what are the big letters B representing? And in subplot C and E, why is there only a single p-value listed?
Comments on the Quality of English LanguageUnfortunately, the quality of the English used in the manuscript makes it very difficult to read. It's not my job to be a copy-editor to improve the level of English. There are many typos, sentence fragments, missing letters, and hard to understand sentences. Please improve the level of English in your manuscript.
Author Response
Reviewer 1
Thank you very much for allowing me to review your manuscript. I found it to be very interesting and potentially important, but I do have a number of questions and comments that I'd like you to address:
- First, your title is very vague. Instead of stating that the transcriptome and proteomic [analysis] revealed the function of ferroptosis, it would be better to state what that function is. Second, I'm not sure what your paper did. What I understand that your paper did was to identify genes and proteins related to ferroptosis that were differentially expressed in the presence or absence of ferroptosis agonists and antagonists. Perhaps stating that would be clearer and correct in the title.
Thank you for your comments, we had changed the title to “Transcriptome and proteome analyses revealed the differences in JEV infected PK-15 cells in response to ferroptosis agonists and antagonists”.
’.
- In line 25-26, in the abstract, you state that "RNA-seq analysis identified 806 differentially expressed genes and 951 differentially expressed proteins in PK-15 cells 24 h after treatment with the ferroptosis agonist and inhibitor, respectively".I'm confused by this sentence. What does the 806 and 951 represent? The number of genes and proteins that were differentially expressed, respectively, when treated with either the agonist or the inhibitor, or the number of genes and proteins when treated with the agonist and inhibitor, respectively?
Thank you for your comments, the number 806 differentially expressed genes was refer to Figure S1A (All the numbers add up) , the number 961 differentially expressed proteins was refer to Figure S3A (All the numbers add up) , we realized the Statistical methods are inappropriate, so we deleted the sentence in line 25-26.
- Could you please add references to your statements in the first paragraph of the introduction?
Thank you for your comments, we had added three references to the first paragraph of the introduction.
- On line 80, you state "After incubating the cells for 2 hrs.." Which cells do you mean, the PK-15 or the BHK-21 cells? Also, what do you mean by MOI=1?
After incubating the cells for 2 hrs, we mean PK-15 cell. The mean of MOI = 1 in virus infection refers to a multiplicity of infection (MOI) where there is, on average, one virus particle per target cell in the experimental system. In this scenario, some cells will be infected with one virus particle, while others may remain uninfected or receive multiple virus particles, depending on the randomness of virus distribution.
- It's unclear on line 81 how the PK-15 cells were infected. Please clarify.
The Jev strain SA14-14-2 (GenBank accession number: AF315119.1) was propagated in BHK-21 cells, then, the PK-15 cells were incubating the JEV solution for 2 h at an MOI = 1, the inoculum was removed, the cells were washed three times with PBS, and fresh medium was added. Infected PK-15 cells were then cultured in MEM medium containing 2% FBS without penicillin/streptomycin.
- In section 2.2, you describe how you used RT-qPCR for 6 specific genes. However, I only understood why you chose these specific genes when I got to page 16 in the Discussion. Please move the first paragraph of page 16 into the introduction so the reader can understand in the methods why you chose these specific genes.
Thank you for your comments, we had moved the first paragraph of page 16 into the introduction.
- On line 92, what does the BCA method mean? Can you briefly describe?
Thank you for your comments, we mean the BCA protein assay kit. The BCA Protein Assay combines the well-known reduction of Cu2+ to Cu1+ by protein in an alkaline medium with the highly sensitive and selective colorimetric detection of the cuprous cation (Cu1+) by bicinchoninic acid (BCA).
8.On line 117, what is the RQN value and why is it important?
RQN (RNA Quality Number) is a numerical score that indicates the quality and integrity of RNA in a sample. It ranges from 1 to 10, where a value of 10 represents highly intact RNA, and lower values indicate increasing levels of RNA degradation. A high RQN indicates good-quality RNA, which is essential for obtaining reliable and consistent data in molecular biology experiments.
9.In your statistical analysis (section 2.2) you state that p<0.05 was considered statistically significant. Is this only for the RT-qPCR data? Did you use any correction for multiple testing, perhaps internally? Figure 4, 5,7, 8 all feature a legend that says "Padjust" that makes me think that the pathway analyses must have adjusted the p-values for multiple testing. Can you confirm that, and state what method they used?
Thank you for your comments, in statistical analysis we state that the p<0.05 was considered statistically significant, that only for RT-qPCR data. For RT-qPCR data, we didn’t use correction for multiple testing. Yes, the pathway analyses must have adjusted the p-values for multiple testing. In transcriptomics and proteomics, Padjust (adjusted p-value) is used to correct for multiple comparisons when performing statistical tests across a large number of genes or proteins. This is crucial because in large-scale studies, such as RNA-seq (transcriptomics) or mass spectrometry (proteomics), testing thousands of genes or proteins simultaneously increases the risk of obtaining false-positive results due to random chance.
10.On line 140, you state that you used One-Way ANOVA for your statistical comparison. I suppose this is for the RT-qPCR analysis in Figure 1. However, even there, I would suspect that you used a two-way ANOVA, because there are two factors: infection status and time (24 h vs 48 h).
Thank you for your comments, the student’s T-test was used to compare two groups at 24 h and 48 h respectively. We had revised the wrong statement.
11.I was surprised to see that the RT-qPCR analysis was only conducted to compare gene expression for the samples with or without JEV infection, but not in the presence of the agonist or antagonist. Why is that? I expected to see the results in Figure 1 in the case of treatments with both agonist and antagonist.
Thank you for your comments, at first, we only want to detect the effects of JEV on ferroptosis related genes, so we didn’t use the agonist or antagonist, and then use the ferroptosis agonist or antagonist to study the influences of ferroptosis on JEV proliferation. The RNA-seq can provide the data of treatments with agonist and antagonist. The data can be found in Table S2.
12.In your RNA sequencing analysis, you compare the differential gene and protein expression in the Negative control group to the same with the treatment with the agonist and the antagonist. Is the Negative Control the same as in Figure 1 where the cells have not been infected with JEV? If that is correct, then all the differences you see between NC and the treatment groups are a combination of changes due to JEV infection and the treatment, and the two are confounded, so it's not possible to separate the two. Did you perhaps mean as Negative Control group cells that have been infected with JEV, but not treated with either the agonist or the antagonist? That would make more sense, and then the results you show are correctly showing the effect of the agonist and the antagonist. However, then you shouldn't call it Negative Control, but call it just JEV. Which one of these is the correct interpretation? I hope that it is the later, since you state on line 198 that JEV-infected PK-15 cells begin to die at 48 hours. I have a similar question about the proteomics analysis. Please clarify!
Thank you for your comments, Negative controls in the transcriptome and proteomic analysis were those cells that were infected with JEV but were not treated with either agonist or antagonist.
13.In Figure 3, and page 6 and 7 you talk about the many differentially expressed genes at 24 and 48 hrs in the different groups. Can you clarify what proportion of the genes differentially expressed at 48 hr were a subset of the genes differentially expressed at 24 hr, so consistent between the timepoints? I have a similar question about Figure 6.
The Expression trend of some DEGs and DEPs in 24 h and 48 h are similar. But some DEGs and DEPs exhibited a different expression trend at 48 hours compared with 24 hours, that may be due to the 48 h is a key timepoint for initiating cytopathic effect (CPE) in JEV infected PK-15 cells. The data can be found in Figure S1, Figure S4, Table S3 and Table S4.
14.On line 214, you introduce the EggNOG classification. Can you explain what that classification is?
EggNOG (Evolutionary genealogy of genes: Non-supervised Orthologous Groups) is a database and bioinformatics tool used for large-scale functional annotation of proteins. It provides orthologs and functional annotation across numerous species and taxonomic levels.
Orthologous Groups (OGs): EggNOG clusters proteins into orthologous groups, representing genes from different species that descended from a common ancestor. These groups are useful for studying evolutionary relationships and predicting gene function.
Functional Annotation: EggNOG assigns functional annotations to proteins, including Gene Ontology (GO) terms, KEGG pathways, and COG (Clusters of Orthologous Groups of proteins) classifications.
Taxonomic Levels: EggNOG provides orthologous groups at different taxonomic levels (e.g., Eukaryota, Bacteria, Archaea). This enables analysis across different scopes of evolutionary divergence.
Phylogenetic Analysis: EggNOG offers tools for phylogenetic analysis to trace the evolutionary history of genes across species.
- In section 3.4, you look at the Gene Expression Pattern analysis, you seem to be combining the results from 24 hrs and 48 hrs in each of the treatments. Is that justified? Are they consistent enough to do that?
Thank you for your advice, we had revised the inappropriate statements in the section 3.4. Due to the 48 h is a key timepoint for initiating cytopathic effect (CPE) in JEV infected PK-15 cells, so some of the go and KEGG enrichment is not consistent between 24 h and 48 h treatment.
- In Figure 4, you show the results of the pathway enrichment analysis of the DEGs. However, could you also show the direction of the change, whether the genes in an enriched pathway are down- or up-regulated, either by showing the average log-fold change for the DEGs in each pathway, or by using the enrichment plot in GSEA? Otherwise, it's hard to tell what the direction of the impact is on gene expression! Perhaps you could do what you did on Figure 10, but of course just for DEGs. The same would apply for Figure 7 just for DEPs.
The Figure 10 used the method of Gene Set Variation Analysis GSVA, we can’t use the GSVA method to generate Figure 4 and Figure 7. GSVA is a non-parametric unsupervised analysis method, mainly by transforming the expression matrix of genes or proteins between different samples into the expression matrix of genes or protein sets between samples, so as to evaluate whether different gene sets/protein sets are enriched in different samples. The GSVA can calculate the enrichment score (ES) of a specific set in each sample, which reflects the enrichment degree of the functional set in the sample.
However, we provide Table S5 and S7, so the genes details of transcriptome GO and KEGG enrichment and whether the genes down- or up-regulated can be find in Table S5; the proteins details of proteome GO and KEGG enrichment can be find in Table S7.
- On line 249, you state that you performed cluster analysis. How was this performed, with what software? These should really go into the Methods section on statistical analysis. Why did you choose subcluster_1 and subcluster_2 for further analysis? The same applies to Figure 8 and line 355.
For Figure 5, We use RSEM software to construct the map by expressing the quantity index Transcripts Per Kilobase Million (TPM). To identify potential trends among the groups, we performed cluster analysis on all DEGs, which were divided into eight subclusters base on gene expression. Since the effects of SP600125 and Erastin on ferroptosis are contrary, we focused on genes that exhibited different expression pattern between the SP600125 and Erastin groups, so we choose subcluster_1 and subcluster_2. For Figure 5, Cluster analysis using python package, choose hierarchical clustering, use the package fastcluster.
18.On line 303, I'm not sure where the numbers 425 and 565 come from. Are these the genes that are specific DEPs to the specific treatment? Please clarify!
The number 425 is come from Figure S5 A (201+224), The number 565 is come from Figure S5 B (217+348).
- On lines 305 and 307, you mention DEGs again. Do you mean DEPs here?
Thank you for your advice, we mean DEPs here, and we revised the mistake on the new line 355-356.
- In the last paragraph of page 11, please discuss your results in the order of the subplots of Figure 7, so you don't have to jump around Figure 7A, then 7C, then 7E, then 7B, 7D and 7F.
Thank you for your comments, we had revised the order in the new line 381-397.
- I'm unclear on where the numbers are coming from in the first and second paragraphs of section 3.6 in terms of the numbers of DEGs and DEPs identified. They don't seem to match up with previous figures in an intuitive way.
Now is at 3.7, Because some DEGs related proteins levels had no significant changes can’t be shown in the diagram, as well as some DEPs related mRNA levels had no significant changes can’t be shown in the diagram.
- On line 394, your stat that "31 named DEGs and DEPs associated with the three groups. However, when I add up all the genes that have consistent expression between DEGs and DEPs in each of the three group pairings, that is 1+11+9+10+11=41 genes. Why do you state 31 instead of 41? What am I missing?
Thank you for your comments, because some genes are shared between groups, so there are only 31 named DEGs and DEPs.
- On line 396, you state that the genes "HMOX1 and CXCL8 was positioned at crucial nodes in the network". How was that determined? They certainly don't have the highest connectivity. What metric was used to assess how crucial each node is in the network? That network analysis should also be explained in the methods under statistical analysis.
Thank you for your comments, we changed the statement to HMOX1 and CXCL8 was positioned at important nodes in the network. We add the network analysis in line 172-174.
- On Figure 10, you show the conjoint pathway enrichment analysis of DEGs and DEPs. This one doesn't have any of the pathways highlighted. How do you know which ones to focus on, and mention in the text?
Thank you for your comments, we had add the highlight on the pathways.
- Again, lines 438-478 should go into the Introduction.
Thank you for your comments, we had put the section to line 66-78.
- On lines 493-497, you list a whole bunch of pathways that were enriched "after ferroptosis inhibitor SP600125 or Erastin treated". So, it this just a list of all the pathway that were enriched in one or the other treatment? Wouldn't it be important to also see which were enriched in both treatments, and how the two treatments contrasted in terms of which pathways were enriched, and also whether those pathways were up- or down-regulated? I don't presently see this comparison. The same would apply to the list of pathways on lines 550-551. Are these the pathways represented by both subclusters? How do they change in response to the different treatments?
Thank you for your comments, we had revised the sentences (lines 493-497) in the new line 532-561; we had also revised the sentences (lines 550-551) in the new line 617-625. We provide the data of genes in go enrichment and KEGG pathways at Table S5 (The top 20 GO or KEGG enrichment were marked in red, and the expression details of genes and proteins can be found in Table S5 and Table S7, respectively).
- Finally, can you speculate on whether Erastin, which promotes ferroptosis, could be used to treat JEV, or not really, because it would to cell death that would be worse than the infection?
Erastin, as an agonist for ferroptosis, whether it is suitable for treating Japanese encephalitis is complex and still requires further research. Erastin’s ferroptosis-inducing effects could theoretically counteract viral replication, but cell death remains in CNS neurons is a serious problem. Therefore, exploring the proper concentration to control the balance between virus inhibition and cell ferroptosis is crucial, that will be helpful for the use the Erastin in JEV-associated encephalitis.
- In Figure S1 and S3, what are the EggNOG categories? What do they represent?
The EggNOG categories can be found in Table S3 now.
- In Figure S2, in subplot B, it looks like all the adjusted p-values are not significant. Why are all the Rich factor values 1 for the last 7 pathways? Also, in subplot D, what are the big letters B representing? And in subplot C and E, why is there only a single p-value listed?
Thank you for your comments, in Figure S2 (we add a new Figure S1, last Figure S2 is Figure S3 now) subplot B the data were analyzed by Goatools software, and the Rich factor value of the last 7 pathways was indeed 1. Also, in subplot D the large letter B is redundant and has been removed. And in subplot C and E all p-values for top20 are given as one value, please see below.
Comments on the Quality of English Language
Unfortunately, the quality of the English used in the manuscript makes it very difficult to read. It's not my job to be a copy-editor to improve the level of English. There are many typos, sentence fragments, missing letters, and hard to understand sentences. Please improve the level of English in your manuscript.
The quality of the English in this manuscript had been edited by the native English speakers, please see the certificate as below.

Reviewer 2 Report
Comments and Suggestions for Authors
In this paper the authors compared the differentially expressed genes and proteins between ferroptosis agonist and inhibitor. The authors provided an insight into the relationship between ferroptosis and virus infection, suggested a possible explanation for the role ferroptosis plays in virus infection defense in pigs.
Overall, this paper is informative, and the logical flow is clear. However, in details, it still needs a lot of polishing works.
My major comments are as follows:
1. In each Results section, the authors should draw a conclusion for the reader to understand what and why they did the experiments and analysis in this section. I know that the authors put the conclusions in the Discussion section. But it would be easier for the readers to follow by drawing the conclusions in Results part.
For example, they should move parts of the statements in line 446 to 478 to section 3.1.
2. In section 3.2, they should explain the differences between the two inhibitors Fer-1 and SP600125: why SP600125 increased the virus loading while Fer-1 not? And same question to the differences between the agonists' effects in protein level and mRNA level: why the agonists inhibit JEV mRNA but not protein?
3. The first paragraph of Section 3.4 is confusing. Please double check the sentences.
4. Figure 5A and Line 250, "were divided into eight subclusters". I did not find the subclusters in Figure 5A. I noticed that there were color codes beside the phylogenetical tree. But it seems that there are only 6 colors. Please clarify or make the color code more significant. I would also suggest that use the same color code for Figure 5B (the same suggestion to Figure 8A and 8B).
5. Section 3.4, the authors indicated that 6 out of 8 subclusters fit the expression pattern. But they only did GO/KEGG analysis to Subcluster 1 and 2. Why? Is there anything special in subcluster 1 and 2? Please clarify.
Minor comments:
1. Figure 2D, why is the band in NS3-20 weaker than the others and the authors claimed "not inhibit"? Is it because of some technical issue? If so, please change to a more representative image. Otherwise, please explain.
2. Figure 2E and 2F, the legend key labels are confusing. It makes no sense to paralleling JEV with the chemicals because even the chemical treatment groups were also infected by JEV. Please change "JEV" to "NC".
3. Line 254 and 260, "GO" enrichment analysis.
4. The authors should introduce some basic information about PK-15 cell line. Just move Line 441~442 to the Introduction section.
5. In figure legends or methods section, the authors should indicate the replication numbers (n=?) and the asterisks definition (*: p<?,**: p<?, etc.) for all statistical figures (Fig. 1, Fig. 2E-F)
Author Response
Reviewer 2
In this paper the authors compared the differentially expressed genes and proteins between ferroptosis agonist and inhibitor. The authors provided an insight into the relationship between ferroptosis and virus infection, suggested a possible explanation for the role ferroptosis plays in virus infection defense in pigs.
Overall, this paper is informative, and the logical flow is clear. However, in details, it still needs a lot of polishing works.
My major comments are as follows:
- In each Results section, the authors should draw a conclusion for the reader to understand what and why they did the experiments and analysis in this section. I know that the authors put the conclusions in the Discussion section. But it would be easier for the readers to follow by drawing the conclusions in Results part.For example, they should move parts of the statements in line 446 to 478 to section 3.1.
Thank you for your comments, we had added the conclusions at the end of results.
- In section 3.2, they should explain the differences between the two inhibitors Fer-1 and SP600125: why SP600125 increased the virus loading while Fer-1 not? And same question to the differences between the agonists' effects in protein level and mRNA level: why the agonists inhibit JEV mRNA but not protein?
Thank you for your comments, Fer-1 inhibits ferroptosis via suppressing the accumulate of ROS, SP600125 inhibits ferroptosis through suppressing the JNK pathway. They inhibited cell ferroptosis through different signaling pathways. Fer-1 could slightly increase the protein level and mRNA level of the virus, while SP600125 could obviously increase the protein level and mRNA level of the virus.
At the protein level, the expression of protein decreased after treatment with both the agonist, whereas the effect of Erastin inhibiting the levels of JEV mRNA is much better than that of RSL3, so the effect of Erastin inhibiting the levels of JEV protein level is much better. RSL3 is the inhibitor of GPX4 (ferroptosis agonist), Erastin is the inducer of ROS and iron-dependent signaling, that may be the reason of the different effects on JEV proliferation.
- The first paragraph of Section 3.4 is confusing. Please double check the sentences.
Thank you for your comments, we revised the sentence in the first paragraph of Section 3.4.
- Figure 5A and Line 250, "were divided into eight subclusters". I did not find the subclusters in Figure 5A. I noticed that there were color codes beside the phylogenetical tree. But it seems that there are only 6 colors. Please clarify or make the color code more significant. I would also suggest that use the same color code for Figure 5B (the same suggestion to Figure 8A and 8B).
Thank you for your comments, in Figure 5A, the colors on the left side are dark green, emerald green, dark red, red, brown, purple, blue and orange, a total of 8 colors, we had revised the image according to your advice in Figure 5 and Figure 8.
- Section 3.4, the authors indicated that 6 out of 8 subclusters fit the expression pattern. But they only did GO/KEGG analysis to Subcluster 1 and 2. Why? Is there anything special in subcluster 1 and 2? Please clarify.
To identify potential trends among the groups, we performed cluster analysis on all DEGs, which were divided into eight subclusters base on gene expression. Since the effects of SP600125 and Erastin on ferroptosis are contrary, we focused on genes that exhibited different expression pattern between the SP600125 and Erastin groups, so we choose subcluster_1 and subcluster_2.
Minor comments:
- Figure 2D, why is the band in NS3-20 weaker than the others and the authors claimed "not inhibit"? Is it because of some technical issue? If so, please change to a more representative image. Otherwise, please explain.
Thank you for your comments, for figure 2D, the band in NS3-20 is weaker than others, we state the Erastin can obviously inhibit the levels of JEV-NS3 protein.
- Figure 2E and 2F, the legend key labels are confusing. It makes no sense to paralleling JEV with the chemicals because even the chemical treatment groups were also infected by JEV. Please change "JEV" to "NC".
Thank you for your comments, we had changed "JEV" to "NC".
- Line 254 and 260, "GO" enrichment analysis.
Thank you for your comments, we had revised the word to GO enrichment analysis.
- The authors should introduce some basic information about PK-15 cell line. Just move Line 441~442 to the Introduction section.
Thank you for your advice, we had moved the sentences to the introduction section in line 92-96.
- In figure legends or methods section, the authors should indicate the replication numbers (n=?) and the asterisks definition (*: p<?,**: p<?, etc.) for all statistical figures (Fig. 1, Fig. 2E-F)
Thank you for your comments, in figure legends and methods section, we had added the replication numbers and the asterisks definition.

Reviewer 3 Report
Comments and Suggestions for Authors
This work describes the relationship that JEV has with ferroptosis, also evaluating a transcriptomic as well as phenotypic approach. In my opinion, the work is well written, and the data obtained would be worthy of publication. Below are some of my comments to address: 1) Line 12: Specify that it is a zoonosis. Also, you should link the part about ferroptosis with an additional sentence ("Some pathways, although known, relationship to JEV unknown" or similar). 2) Line 21: Check "ferroptosis." 3) Line 23: Agonist or "inducer"? Please check. 4) Line 40: More information on the virus, spread, etc. would be appropriate. 5) How was the choice of MOI and time made? 6) WB: Are there any uncropped images of the western blots? How do the journal and the editors understand that the different pieces of membrane derive from the same electrophoretic run? 7) Figure 4 and others: too small, the results are poorly appreciated. Line 441: It would be necessary to link the two sentences better. 9) Line 516: I suggest the authors take into consideration other viruses capable of causing apoptosis and activation of the PI3K/Akt pathway from the veterinary world, such as feline herpes virus, PRV.
Author Response
Reviewer 3
This work describes the relationship that JEV has with ferroptosis, also evaluating a transcriptomic as well as phenotypic approach. In my opinion, the work is well written, and the data obtained would be worthy of publication.
Below are some of my comments to address:
- Line 12: Specify that it is a zoonosis. Also, you should link the part about ferroptosis with an additional sentence ("Some pathways, although known, relationship to JEV unknown" or similar).
We had added the sentence “Ferroptosis occurred in encephalitis, but the relationship between JEV and ferroptosis is still unclear” in line 13-14.
- Line 21: Check "ferroptosis."
Thank you for your comments, we had revised the “ferrpotosis” to “ferroptosis” in the new line 23.
- Line 23: Agonist or "inducer"? Please check.
Thank you for your advice, we had changed the words "inducer" to “agonist” in line 25.
4)Line 40: More information on the virus, spread, etc. would be appropriate.
Thank you for your comments, we had added the sentences "JEV is a single-stranded RNA virus with neurotoxic and neuroinvasive characteristics. Functional proteins encoded during JEV replication include the NS3 protein, which has three distinct enzyme activities that contribute to viral replication and assembly. Infection with JEV results in acute central nervous system diseases in humans. JEV infection outcomes in pigs do not vary by breed or sex. JEV infection occurs frequently in pigs in summer" in line 37-42.
5)How was the choice of MOI and time made?
For PK-15 cells, in our pre-experiment, we found the MOI=1 and incubating the JEV solution for 2 h
is a suitable concentration and time, mRNA and protein can be detecting at 12, 24 h, 36 and 48 h, the cells were didn’t initiate cytopathic effect (CPE), and at 60 h after infection, the cell initiate CPE. please see the image below.
6)WB: Are there any uncropped images of the western blots? How do the journal and the editors understand that the different pieces of membrane derive from the same electrophoretic run?
To reduce the use of antibodies, we did not incubate the entire PVDF membrane with antibodies. Instead, each band was cut according the marker (NS3 60-75kD, TBB5 45-60kD) from the PVDF membrane and incubated with the primary antibody individually. As a result, there is no complete marker in the original image. We had performed uncropped images recently, that demonstrated our bands are on the right position. NS3 (Cat: GTX125868, Genetex, USA), TBB5 (Cat:AM1031A, Abgent, China)
7)Figure 4 and others: too small, the results are poorly appreciated.
Thank you for your comments, the PPI of the Figure 4 and others is 300, when enlarge the image the font can be clearly visible. We had enlarger part of the font in the Figures, and we sorry for part of font can’t be enlarged.
8)Line 441: It would be necessary to link the two sentences better.
Thank you for your comments, we had moved the sentences (line 441) to the introduction section in line 92-96.
- Line 516: I suggest the authors take into consideration other viruses capable of causing apoptosis and activation of the PI3K/Akt pathway from the veterinary world, such as feline herpes virus, PRV.
Thank you for your advice, we had revised the sentence in the new line 580-582.

Round 2
Reviewer 1 Report
Comments and Suggestions for Authors
Dear Authors,
Thank you very much for your thorough revision of your manuscript. I do believe that the quality of your manuscript has improved considerably, and I was relieved to read that the control in the transcriptomics/proteomics studies were JEV-infected cells. I do still have one big, and some smaller comments and concerns, please see them below:
The major question I have is whether or not the genes and proteins that you found upregulated by the ferroptosis agonists are positively associated with ferroptosis, and those that you found downregulated by the ferroptosis agonists, were they negatively associated with ferroptosis? Conversely, were the genes and proteins downregulated by the ferroptosis antagonist, and upregulated by the ferroptosis antagonist, were they negatively associated with ferroptosis? That is not clear to me from your analysis. I'm generally confused why you did not do enrichment analysis on the up- and down-regulated genes/proteins separately, to see which pathways are down- or up-regulated. Your results are still useful as they are, they are just incomplete because of that.
Minor comments:
1. On line 25, please put "experimentally infected" before "PK15 cells".
2. On line 41, you state that "JEV infection outcomes in pigs do not vary by breed and sex.". Is that correct? In the next two sentences, you talk about the differential impact of JEV infection in sows and boars.
3. In your title, remove "the" before "differences", and put a dash between "JEV" and "infected".
4. On line 46-47, you state that "ferroptosis is an emerging form of regulated cell death". How is it "emerging"? You constrast it with "traditional" forms of cell death. This sounds strange. Do you mean that ferroptosis is a form of cell death that we're just starting to recognize, while we've known for a long time about the other forms of cell death? If yes, then say that.
5. On line 83, you talk about how ferroptosis is generally considered a stimulatory factor for viral replication. However, in your study, JEV infection led to ferroptosis, which leads to cell death, and increased ferroptosis led to lower JEV levels. Many of the pathways you found differentially expressed were related to the immune response. Can we argue that ferroptosis is part of the immune response, and is responsible for killing infected cells to restrict the proliferation of JEV? Ultimately, the virus can't replicate in a dead cell. So how is then ferroptosis stimulatory for viral replication? This is an apparent contradiction for me. Can you please explain how ferroptosis benefits viral replication?
6. On line 91, please add "JEV-infected" before "PK-15 cells.
7. In the Introduction, can you please also detail what is already known about the relationship between ferroptosis and JEV-infection in these cell lines. If this is completely novel, please state that as well.
8. On line 104, please put a space before "with".
9. Line 105-106 seems to be a sentence fragment.
10. On line 108, please put "in" after "incubating".
11. On line 113, please remove comma after "PTGS2".
12. On line 117, please start a new sentence after "Takara)".
13. On line 135, please remove space between "Image" and "J".
14. On line 152, please start new sentence at "clean".
15. On line 155 and on line 165, please add "adjusted" before "P-value".
16. Do I understand correctly that you did not correct for multiple testing either in the analysis in Figure 1 or Figure 2? If you do pairwise t-tests between the JEV and the control for each measurements for both timepoints on Figure 2, that is 18 comparisons. If you compare the three treatment at both timepoints, that is 10 more tests. The family-wise Type-I error rates for a significance level of 0.05 is >40%, which is very high. I'd like you to at least address this in your discussion, even if you choose not to correct for that.
17. In the last paragraph of section 3.3, you talk about the different EggNOG pathways that the agonist and the antagonist enriched. You present them as very different by saying "In contrast,". However, they are actually the same pathways, except for "chromatic structure and dynamics". Can we rephrase this section to highlight the similarities as well, and then pull out the difference?
18. On page 9, you talk about the cluster analysis of the DEGs. You specifically do the enrichment analysis for subcluster_1 and subcluster_2. You don't detail here why you chose these two subclusters. However, in your response to reviewers, you revealed that you chose these because they have opposite patterns. Please state that in the text as well.
19. On line 324, please add "characteristics" after "quantitative".
20. On the top of page 12, in the first paragraph, you detail the number of unique DEPs associated with the antagonist and agonist. I do understand now where your numbers are coming from, but for the reader, it would be benefitial if you made it clear that you're talking about the unique DEPs. You should repeat the sentence you have on the top of page 8 about the commonly expressed DEGs (but of course here with DEPs), and state that you're focusing on the unique ones.
21. On line 453, you state that "HMOX1 and CXCL8 were positioned as important nodes in the network". What do you mean by "important", in what way are they important? How was this determined?
22. On line 502, please change "though" to "through".
23. On line 503, please remove "be".
24. On line 517, please change "better" to "higher".
25. On line 519, please change "ti" to "to".
26. On line 572, please start a new sentence at "IL-17A".
27. On line 573, please change "mainly" to "main".
28. On line 574, please start a new sentence at "Elevated".
29. On line 618, please change "reult" to "result".
30. The sentence starting on line 618 as "In which.." is a fragment.
31. On line 630, please change "increases" to "increase".
32. On line 649, please change "ZIKVA" to just "ZIKV", if you just refer to Zika virus.
33. On line 656, please remove "can".
34. On line 668, please remove "Erastin's".
Author Response
Thank you very much for your thorough revision of your manuscript. I do believe that the quality of your manuscript has improved considerably, and I was relieved to read that the control in the transcriptomics/proteomics studies were JEV-infected cells.
Thank you very much for your carefully review our manuscript and help us improving the quality of the manuscript.
I do still have one big, and some smaller comments and concerns, please see them below:
The major question I have is whether or not the genes and proteins that you found upregulated by the ferroptosis agonists are positively associated with ferroptosis, and those that you found downregulated by the ferroptosis agonists, were they negatively associated with ferroptosis? Conversely, were the genes and proteins downregulated by the ferroptosis antagonist, and upregulated by the ferroptosis antagonist, were they negatively associated with ferroptosis? That is not clear to me from your analysis. I'm generally confused why you did not do enrichment analysis on the up- and down-regulated genes/proteins separately, to see which pathways are down- or up-regulated. Your results are still useful as they are, they are just incomplete because of that.
Thank you for your comments, we had performed the upregulated and downregulated DEGs or DEPs analysis according your guidance, and some of the upregulated DEGs and DEPs by the Erastin are associated with ferroptosis, but not downregulated DEGs. We also provided Figure S4, Figure S5, Figure S9, and Figure S10 of the enrichment analysis on the up- and down-regulated genes/proteins separately. The related statements were on line 308-342 and 456-486 .
Minor comments:
- On line 25, please put "experimentally infected" before "PK15 cells".
Thank you for your comments, we had added "experimentally infected" before "PK15 cells" on line 25.
- On line 41, you state that "JEV infection outcomes in pigs do not vary by breed and sex.". Is that correct? In the next two sentences, you talk about the differential impact of JEV infection in sows and boars.
Thank you for your comments, we had revised the statement to "JEV infection outcomes in pigs do not vary by breed" on line 42.
- In your title, remove "the" before "differences", and put a dash between "JEV" and "infected".
Thank you for your comments, we had revised the title as your advice.
- On line 46-47, you state that "ferroptosis is an emerging form of regulated cell death". How is it "emerging"? You constrast it with "traditional" forms of cell death. This sounds strange. Do you mean that ferroptosis is a form of cell death that we're just starting to recognize, while we've known for a long time about the other forms of cell death? If yes, then say that.
Thank you for your comments, we had deleted the inappropriate word "emerging" on line 47.
- On line 83, you talk about how ferroptosis is generally considered a stimulatory factor for viral replication. However, in your study, JEV infection led to ferroptosis, which leads to cell death, and increased ferroptosis led to lower JEV levels. Many of the pathways you found differentially expressed were related to the immune response. Can we argue that ferroptosis is part of the immune response, and is responsible for killing infected cells to restrict the proliferation of JEV? Ultimately, the virus can't replicate in a dead cell. So how is then ferroptosis stimulatory for viral replication? This is an apparent contradiction for me. Can you please explain how ferroptosis benefits viral replication?
Ferroptosis often presents a double-edged sword in the context of viral infections; while it is a mechanism of cell death that can limit viral spread, it can also be co-opted by some viruses to enhance their replication and survival. Iron is crucial for various cellular functions, including DNA synthesis and metabolism. Viruses can manipulate iron metabolism to enhance their replication. Increased iron levels can lead to ferroptosis, which may create a favorable environment for viral propagation by disrupting host cell functions and promoting inflammation (Jing Wang, Front Microbiol, 2023)
- On line 91, please add "JEV-infected" before "PK-15 cells.
Thank you for your comments, we had added the words "JEV-infected" before "PK-15 cells on line 96.
- In the Introduction, can you please also detail what is already known about the relationship between ferroptosis and JEV-infection in these cell lines. If this is completely novel, please state that as well.
Thank you for your comments, we had added the sentence of the relationship between ferroptosis and JEV-infection on line 89-91.
- On line 104, please put a space before "with".
Thank you, we had put a space before "with" on line 109.
- Line 105-106 seems to be a sentence fragment.
Thank you, we had revised the sentence on line 110-113.
- On line 108, please put "in" after "incubating".
Thank you, we had put "in" after "incubating" on line 114.
- On line 113, please remove comma after "PTGS2".
Thank you, we had deleted the comma after "PTGS2"on line 119.
- On line 117, please start a new sentence after "Takara)".
Thank you, we had started a new sentence after "Takara)" on line 123.
- On line 135, please remove space between "Image" and "J".
Thank you, we had removed space between "Image" and "J" on line 143.
- On line 152, please start new sentence at "clean".
Thank you, we had started a new sentence at “clean” on line 161.
- On line 155 and on line 165, please add "adjusted" before "P-value".
Thank you, we had added "adjusted" before "P-value" on line 164 and175.
- Do I understand correctly that you did not correct for multiple testing either in the analysis in Figure 1 or Figure 2? If you do pairwise t-tests between the JEV and the control for each measurements for both timepoints on Figure 2, that is 18 comparisons. If you compare the three treatment at both timepoints, that is 10 more tests. The family-wise Type-I error rates for a significance level of 0.05 is >40%, which is very high. I'd like you to at least address this in your discussion, even if you choose not to correct for that.
Sorry, we made a wrong statement, in figure 1 we used T-tests, in figure 2 we used ANOVA analysis to the RT-qPCR results, we had revised the statement in the section 2.7.
- In the last paragraph of section 3.3, you talk about the different EggNOG pathways that the agonist and the antagonist enriched. You present them as very different by saying "In contrast,". However, they are actually the same pathways, except for "chromatic structure and dynamics". Can we rephrase this section to highlight the similarities as well, and then pull out the difference?
Thank you for your advice, we had changed the words "In contrast" to "Similarly" on line 271.
18.On page 9, you talk about the cluster analysis of the DEGs. You specifically do the enrichment analysis for subcluster_1 and subcluster_2. You don't detail here why you chose these two subclusters. However, in your response to reviewers, you revealed that you chose these because they have opposite patterns. Please state that in the text as well.
Thank you for your comments, we had added relative sentence on line 352-354 .
- On line 324, please add "characteristics" after "quantitative".
Thank you, we had added "characteristics" after "quantitative" on line 379.
- On the top of page 12, in the first paragraph, you detail the number of unique DEPs associated with the antagonist and agonist. I do understand now where your numbers are coming from, but for the reader, it would be benefitial if you made it clear that you're talking about the unique DEPs. You should repeat the sentence you have on the top of page 8 about the commonly expressed DEGs (but of course here with DEPs), and state that you're focusing on the unique ones.
Thank you for your advice, we had talking about the unique DEGs and DEPs on page 8 (line 263-264) and page 12 (409-410).
- On line 453, you state that "HMOX1 and CXCL8 were positioned as important nodes in the network". What do you mean by "important", in what way are they important? How was this determined?
Thank you for your comments, we realized the statement is inappropriate, we had changed the sentence as " we found HMOX1, CXCL8 were positioned in the network, which were interact with ferroptosis related genes GCLM or F2R respectively" on line 542-543.
- On line 502, please change "though" to "through".
Thank you, we had changed "though" to "through" on line 592.
- On line 503, please remove "be".
Thank you, we had removed "be" between could and serve on line 594.
- On line 517, please change "better" to "higher".
Thank you, we had changed "better" to "higher" on line 609.
- On line 519, please change "ti" to "to".
Thank you, we had changed "ti" to "to" on line 611.
- On line 572, please start a new sentence at "IL-17A".
Thank you, we had started a new sentence at "IL-17A" on line 665.
- On line 573, please change "mainly" to "main".
Thank you, we had changed "mainly" to "main" on line 666.
- On line 574, please start a new sentence at "Elevated".
Thank you, we had started a new sentence at "Elevated" on line 667.
- On line 618, please change "reult" to "result".
Thank you, we had deleted the words "as a reult" before “Among these” on line 715.
- The sentence starting on line 618 as "In which.." is a fragment.
Thank you, we had changed "in which" to "among these pathways" on line 715.
- On line 630, please change "increases" to "increase".
Thank you, we had changed "increases" to "increase" on line 729.
- On line 649, please change "ZIKVA" to just "ZIKV", if you just refer to Zika virus.
Thank you, we had changed "ZIKVA" to just "ZIKV" on line 748.
- On line 656, please remove "can".
Thank you, we had removed "can" between “48 h” and “elevated” on line 755.
- On line 668, please remove "Erastin's".
Thank you, we had removed "Erastin's" between “the” and “inhibitory” on line 769.
Reviewer 2 Report
Comments and Suggestions for Authors
The authors have addressed all my concerns.
Author Response
The authors have addressed all my concerns.
Thank you very much for your guidance and help us improving the quality of the manuscript.
Reviewer 3 Report
Comments and Suggestions for Authors
The authors have addressed a great part of my previous comments. I have only minor comments:
1) Responses to my previous point 5 and 6 should be provided in the main text in the material and methods section (or elsewhere).
2) The authors have provided works about PRV and PI3K/Akt/mTOR axis but not about FeHV-1.
Author Response
The authors have addressed a great part of my previous comments. I have only minor comments:
- Responses to my previous point 5 and 6 should be provided in the main text in the material and methods section (or elsewhere).
Thank you for your advice, we had add the statement on line 188-191.
2) The authors have provided works about PRV and PI3K/Akt/mTOR axis but not about FeHV-1.
Thank you for your advice, we had revised the sentence in the new line 676-678.